# CSF1R inhibitors induce a sex-specific resilient microglial phenotype and functional rescue in a tauopathy mouse model

Noah R. Johnson [1,2,3,7], Peng Yuan [1,4,7], Erika Castillo [1], T. Peter Lopez[1], Weizhou Yue [1], Annalise Bond [1], Brianna M. Rivera[1], Miranda C. Sullivan[1], Masakazu Hirouchi[1,5], Kurt Giles[1,6], Atsushi Aoyagi[1,5] & Carlo Condello [1,6,7] ✉

Microglia are central to pathogenesis in many neurological conditions. Drugs targeting colony-stimulating factor-1 receptor (CSF1R) to block microglial proliferation in preclinical disease models have shown mixed outcomes, thus the therapeutic potential of this approach remains unclear. Here, we show that CSF1R inhibitors given by multiple dosing paradigms in the Tg2541 tauopathy mouse model cause a sex-independent reduction in pathogenic tau and reversion of non-microglial gene expression patterns toward a normal wild type signature. Despite greater drug exposure in male mice, only female mice have functional rescue and extended survival. A dose-dependent upregulation of immediate early genes and neurotransmitter dysregulation are observed in the brains of male mice only, indicating that excitotoxicity may preclude functional benefits. Drug-resilient microglia in male mice exhibit morphological and gene expression patterns consistent with increased neuroinflammatory signaling, suggesting a mechanistic basis for sex-specific excitotoxicity. Complete microglial ablation is neither required nor desirable for neuroprotection and therapeutics targeting microglia must consider sex-dependent effects.

Microglia, the resident innate immune cells of the central nervous system (CNS), are important for neurodevelopment and homeostasis, and are a fundamental component to pathogenesis in many neurological conditions. We now appreciate that microglia are heterogeneous cells, are influenced by the periphery, have sex-dependent biology, and can be helpful or harmful depending on the disease stage or specific pathology[1–4]. Gene mutations affecting the expression and sequence of microglial genes (e.g. *TREM2*, *CD33*, and *MS4A*) increase risk for Alzheimer's disease (AD), and implicate microglia in several disease pathways including toxic protein aggregation (Aβ and tau) and neuroinflammation[5,6]. Thus, for the first time, there is unequivocal evidence in humans that certain microglial functions are robustly

[1]Institute for Neurodegenerative Diseases, UCSF Weill Institute for Neurosciences, University of California, San Francisco, CA 94158, USA. [2]University of Colorado Alzheimer's and Cognition Center, Department of Neurology, University of Colorado Anschutz Medical Campus, Aurora, CO 80045, USA. [3]Linda Crnic Institute for Down Syndrome, University of Colorado Anschutz Medical Campus, Aurora, CO 80045, USA. [4]Department of Rehabilitation Medicine, Huashan Hospital, State Key Laboratory of Medical Neurobiology, Institute for Translational Brain Research, MOE Frontiers Center for Brain Science, Fudan University, Shanghai 200032, China. [5]Daiichi Sankyo Co., Ltd., Tokyo 140-8710, Japan. [6]Department of Neurology, UCSF Weill Institute for Neurosciences, University of California, San Francisco, CA 94158, USA. [7]These authors contributed equally: Noah R. Johnson, Peng Yuan, Carlo Condello. ✉e-mail: carlo.condello@ucsf.edu

involved in the pathogenesis of neurodegenerative disease. However, the precise mechanisms governing microglia function in disease are still not well understood.

In tauopathy (a family of neurodegenerative disorders characterized by tau inclusions in neural cells), there is growing evidence that microglia play an early and constant role in tau aggregation and neuronal loss. Disease-activated microglia can secrete pro-inflammatory cytokines that regulate neuronal kinases and phosphatases causing tau hyperphosphorylation, aggregation and consequent neurodegeneration[7–9]. Genome-wide transcriptomic studies have identified innate immune pathways that implicate early and robust involvement of microglia in human tauopathy[10,11] and related mouse models[12,13]. Deletion of microglial-specific genes or genetic ablation of microglial cells in rodents have been useful approaches to dissect microglial-mediated mechanisms in disease models, but pharmacologic tools to more dynamically manipulate microglial function have been limited. Recently developed small-molecule drugs targeting colony-stimulating factor-1 receptor (CSF1R), a receptor kinase critical for survival and proliferation of CNS microglia, peripheral tissue macrophages and blood myeloid cells[14], are approved for clinical use in various oncology indications[15], and have now been adopted by the neuroscience community to study microglial biology. In the past few years, there have been numerous studies using CSF1R inhibitors in models of neurological disease, but only a few studies in models of primary tauopathy[16–19]. While important first steps, these studies only explored a single, static time point of treatment, or used only one sex. Given the dynamic nature and complexity of microglial activation, the timing of CSF1R inhibition in tauopathy and its translational relevance is still an open question.

Thus, the goal of our study was to define a therapeutic window that not only reduced pathological markers, but also led to functional improvement. Moreover, we questioned whether complete or continuous microglial ablation using CSF1R inhibitors was necessary, given the important and diverse roles these cells play in brain health and disease. Here, we systematically tested CSF1R inhibition using multiple drug analogs at several time points in transgenic mice developing spontaneous tauopathy, and in an inoculation model of induced tauopathy. We demonstrated a reduction of tau pathology in multiple dosing schemes without complete microglial ablation; drug exposure levels were correlated with the extent of tau-prion[20] and microglial reduction. Unexpectedly, we observed suppressed plasma biomarkers of neurodegeneration, rescue of aberrant behavior, and extended survival in female mice only. These data reveal a previously unrecognized sex-dependent therapeutic benefit of pharmacological CSF1R inhibition. Transcriptome analyses showed that treated tauopathy mice exhibited a restored gene expression profile similar to wild type mice; however, we observed a specific module of sex- and drug concentration-dependent gene expression that might explain the lack of functional rescue in male mice. Residual microglia had a morphology similar to wild type microglia and their gene expression pattern indicated a unique, sex-specific signature in response to CSF1R inhibition. These data highlight yet another context for microglial heterogeneity with implications for understanding microglial biology, and argue that tempering microglial activation with drugs, rather than microglial ablation, is a better therapeutic strategy with clinical relevance.

## Results

### CSF1R inhibition reduces pathogenic tau in the brains of Tg2541 mice

Building on previous findings[16–19], we first evaluated the effect of CSF1R inhibition on the levels of pathogenic tau in the brains of transgenic mice expressing human tau, using a cell-based tau-prion bioassay, enzyme-linked immunosorbent assay (ELISA), and immunohistochemical (IHC) analysis. To deplete microglia, Tg2541 mice were dosed with one of two potent, orally bioavailable, and brain-penetrant CSF1R inhibitors: PLX3397 (pexidartinib), which binds receptor tyrosine kinases CSF1R, and to lesser extent, KIT and FLT3[21], and PLX5622, which selectively binds CSF1R[22]. Three different treatment paradigms were evaluated: acute (2–4 months old), chronic (2–7 months old), and terminal (2 months old until death) (Fig. 1a–c). Transgenic B6-Tg(Thy1-MAPT*P301S)2541 mice[23], referred to here as Tg2541 mice, express the 0N4R isoform of human tau with the familial frontotemporal lobar degeneration (FTLD)-linked P301S mutation[24], which increases its aggregation propensity and prion-like characteristics[25,26]. We previously demonstrated that the levels of pathogenic tau in hindbrain regions of Tg2541 mice were greater than in forebrain regions[27]. This observation is consistent with the neuropathological staging of human FTLD-tau and specifically of progressive supranuclear palsy (PSP) where tau deposition begins and predominates in subcortical and brainstem nuclei[28]. Therefore, the forebrain and hindbrain regions were examined separately in this study (Fig. 1d).

We confirmed that CSF1R inhibition effectively reduced microglial markers Iba1 and P2yr12 by an average of ~60% in both the forebrains and hindbrains of Tg2541 mice compared to vehicle treatment, and that they had similar effects in the brains of C57BL/6 J wild type mice (Supplementary Fig. 1a–p). Principal component analysis of all Iba1 and P2yr12 data combined showed that sex did not have a significant effect on the extent of microglial depletion by CSF1R inhibitors (Supplementary Fig. 1q and Supplementary Data File 1), and therefore male and female mice were grouped together for analysis unless otherwise noted.

We next employed a reproducible and rapid cell-based bioassay[29,30] to measure the activity of replication-competent tau-prions in brain homogenates from Tg2541 mice. To ensure an appropriate dynamic range in this bioassay, we optimized the dilution factor and assay duration using aged Tg2541 mouse brain samples, which showed greater than 100-fold higher signal than wild type mouse brain samples (Supplementary Fig. 2). Following acute, chronic, or terminal treatment with PLX3397 or PLX5622, tau-prion activity in the forebrains of Tg2541 mice was significantly decreased compared to vehicle-treated mice (Fig. 1e–g). Hyperphosphorylation and aggregation of tau in Tg2541 mice occurs first in hindbrain regions, especially in the brainstem and spinal cord, leading to motor deficits causing severe paraparesis[23]. This is consistent with our previous report of early and aggressive tau-prion activity in hindbrain regions of Tg2541 mice[27]; as such, we found that acute CSF1R inhibition was insufficient to reduce tau-prion activity in the hindbrain (Fig. 1e). However, chronic or terminal treatment with PLX3397 did significantly reduce tau-prion activity in hindbrain regions (Fig. 1f, g) and the spinal cords of Tg2541 mice (Supplementary Fig. 3). To examine other markers of pathogenic tau, we measured the levels of tau phosphorylated at Ser396 (pS396) by ELISA, and tau phosphorylated at Ser202/Thr205 (pS202/T205) by IHC. Acute, chronic, or terminal PLX3397 treatment robustly reduced pS396 tau in both forebrain and hindbrain regions of Tg2541 mice (Fig. 1h–j), and also reduced pS202/T205 tau in forebrain regions (Fig. 1k–m) and in the spinal cord (Supplementary Fig. 3d). Since the various measures of tau pathology represent different steps of tau pathogenesis (hyperphosphorylation vs. oligomerization vs. filament formation), they may be differentially impacted by CSF1R inhibition with different treatment regimens. Thus, to consider all tau measurements and both brain regions together, we performed principal component analysis which revealed that pathogenic tau was reduced by both CSF1R inhibitors, and that there was no significant effect of sex on drug efficacy (Fig. 1n and Supplementary Data File 1).

Having verified the benefits of microglial depletion at an early disease stage, we next wondered whether initiating CSF1R inhibition at a more advanced stage of disease would have similar effects, simulating an interventional drug treatment. Thus, we dosed Tg2541 mice with PLX3397 in a delayed treatment paradigm (4–7 months old). Similar to

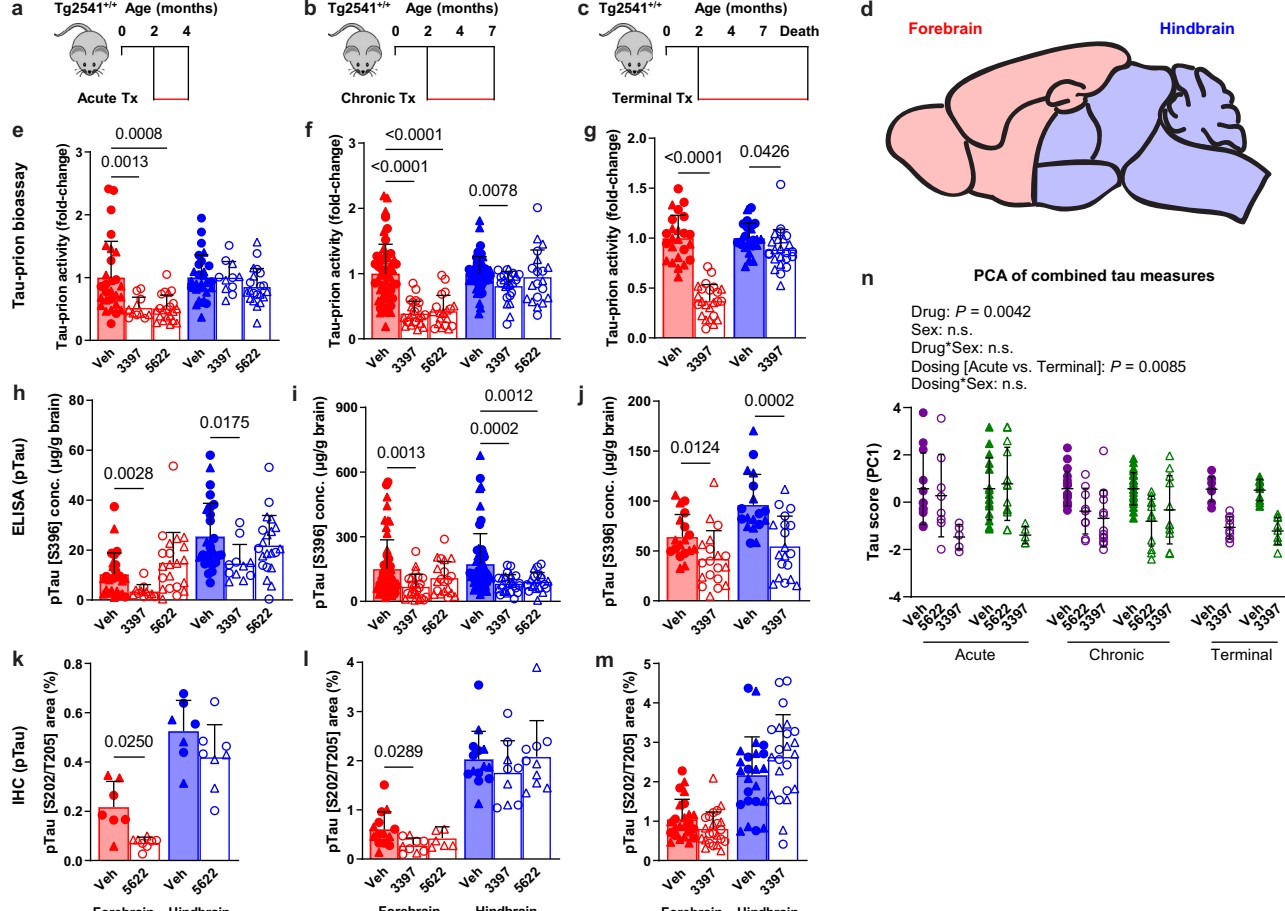

**Fig. 1 | CSF1R inhibition by three treatment paradigms reduces pathogenic tau levels in the brains of Tg2541 mice. a–c** Schematics of acute (**a**), chronic (**b**), or terminal (**c**) PLX treatment of Tg2541 mice from 2–4 mo of age, 2–7 mo of age, or 2 mo of age until death, respectively. **d** Sagittal view of the mouse brain divided into two regions: the forebrain, containing the cortex, hippocampus, striatum, and olfactory bulb; and the hindbrain, containing the thalamus, hypothalamus, midbrain, cerebellum, and brain stem. **e–g** Tau-prion levels in forebrain and hindbrain tissue homogenates of Tg2541 mice receiving acute (**e**, n = 28 vehicle-treated mice, 11 PLX3397-treated mice, and 21 PLX5622-treated mice), chronic (**f**, n = 53 vehicle-treated mice, 24 PLX3397-treated mice, and 20 PLX5622-treated mice), or terminal (**g**, n = 24 vehicle-treated mice and 24 PLX3397-treated mice) treatment with vehicle, PLX3397 (275 mg/kg oral), or PLX5622 (1200 mg/kg oral), measured using the HEK293T cell tau-prion bioassay and normalized to the vehicle-treated group. **h–j** Levels of pTau [S396] measured by ELISA in formic acid extracts of forebrain and hindbrain tissue homogenates of Tg2541 mice receiving acute (**h**), chronic (**i**), or terminal (**j**) treatment with vehicle, PLX3397, or PLX5622, normalized to total protein concentration. Similar numbers of mice were used as in (**e–g**). **k–m** Quantification of pTau [S202/T205]-positive area by IHC analysis of forebrain and hindbrain areas of Tg2541 mice receiving acute (**k**, n = 7 vehicle-treated mice and 8 PLX5622-treated mice), chronic (**l**, n = 14 vehicle-treated mice, 9 PLX3397-treated mice, and 10 PLX5622-treated mice), or terminal (**m**, n = 24 vehicle-treated mice and 23 PLX3397-treated mice) treatment with vehicle, PLX3397, or PLX5622.

Welch ANOVA with Dunnett T3 post hoc testing was used in (**e**, **f**, **h**, **i**, and **l**). Two-way ANOVA with Holm–Šidák post hoc testing was used in (**g**, **j**, **k**, and **m**). P values for all statistically significant differences (P < 0.05) are shown. **n**, Principal component analysis was performed, using all tau-prion and pTau[S396] data presented in (**e–j**) to calculate a 'tau score' that represents the amount of pathogenic tau in both the forebrains and hindbrains of Tg2541 mice. All data was first standardized to the respective vehicle-treated group of the same sex and same dosing paradigm. Then, two principal components (PC1 and PC2) were identified which accounted for 70.7% of the total variance in the data. Multiple linear regression was performed on PC1 of the drug-treated groups to evaluate the main effects of sex and dosing paradigm, and the dosing*sex interaction effect. Multiple linear regression was performed on all groups to determine the main effect of drug and the drug*sex interaction effect. P values for all statistically significant differences (P < 0.05) are shown. n.s. indicates not statistically significant. PC2 was also evaluated, but only the drug main effect was statistically significant. n = 14 acute vehicle, 17 acute PLX3397, 18 acute PLX5622, 28 chronic vehicle, 35 chronic PLX3397, 22 chronic PLX5622, 18 terminal vehicle, and 19 terminal PLX3397-treated mice. In (**e–n**), each symbol represents the forebrain or hindbrain of an individual mouse, with female mice shown as closed or open circles and male mice shown as closed or open triangles. Error bars represent the s.d. of the mean. Source data are provided as a Source Data file.

terminal treatment, interventional treatment also significantly reduced tau-prion activity in both the forebrain and hindbrain (Supplementary Fig. 4a, b). Although pS396 tau levels were unchanged after interventional treatment, the levels of tau phosphorylated at Thr231 (pT231) were reduced in the forebrain (Supplementary Fig. 4c, d). Considering the potential off-target effects of continuous, long-term microglial depletion on brain function, we also wondered whether periodic CSF1R inhibition might provide a safer, yet similarly efficacious therapy. Thus, we also tested PLX3397 dosed intermittently by repeating dosing cycles of three weeks on, followed by three weeks off. The intermittent

dosing interval was selected based on prior studies showing that there is rapid microglial repopulation of the brain, and that morphological and transcriptional microglial changes return to baseline levels within 21 days of removing PLX[31]. Intermittent treatment produced similar reductions in the levels of microglial markers in both brain regions as for continuous treatment (Supplementary Fig. 1r–t), but tau-prion activity and pT231 levels were reduced only in the forebrain (Supplementary Fig. 4e–h). Taken together, these data suggest that intermittent/interventional dosing is sufficient to reduce pathogenic tau in the forebrain of Tg2541 mice, likely due to slower disease kinetics;

however, continuous CSF1R inhibition is necessary for the extended reduction of pathogenic tau in the hindbrain.

Tau has been shown to propagate throughout the brain in a prion-like fashion along interconnected neural networks[32,33]. To test the hypothesis that microglial depletion may reduce the propagation of tau-prions[20] in the brains of Tg2541 mice, we inoculated fibrils of the microtubule-binding repeat domain of tau, referred to as K18 fibrils[34], into the hippocampus and overlying cortex (forebrain regions) of Tg2541 mice and then treated them with PLX3397. Compared to un-inoculated mice, K18-inoculated mice had significantly increased tau-prion levels in the ipsilateral (inoculated) forebrain, as well as in the contralateral forebrain and in the hindbrain (Supplementary Fig. 5), which suggests that tau-prions had propagated from the inoculation site to those brain regions. However, acute PLX treatment was sufficient to significantly reduce tau-prion levels in the ipsilateral forebrain and hindbrain, as well as in the contralateral forebrain. Furthermore, tau-prion levels in the contralateral forebrain of the inoculated PLX-treated mice were not significantly different from the forebrain of un-inoculated, vehicle-treated mice (Supplementary Fig. 5), which indicates that CSF1R inhibition prevented the spreading of tau-prions from the inoculation site to those brain regions.

CSF1R inhibition can affect peripheral immune cells such as blood myeloid cells and tissue macrophages, in addition to microglia[17,35]. To determine if the effects of PLX3397 and PLX5622 on pathogenic tau in the brain were due, in part, to depletion of peripheral CSF1R-expressing cells we dosed Tg2541 mice with PLX73086[36], a non-brain penetrant CSF1R inhibitor analog of PLX3397 and PLX5622. Chronic treatment with PLX73086 had no significant effect on microglial markers Iba1 and P2yr12, or on levels of tau-prions, pTau[S396], or pTau[T231] in the forebrains or hindbrains of Tg2541 mice (Supplementary Fig. 6). Therefore, the effects of CSF1R inhibitors in peripheral compartments do not significantly contribute to their reduction of pathogenic tau in the CNS. We also evaluated the numbers of Iba1+/CD206+ perivascular macrophages (PVMs) and found that PLX3397 treatment did not significantly deplete this cell population, although there was a trend ($P = 0.0947$) towards reduced PVMs in female Tg2541 mice (Supplementary Fig. 7). Lastly, because there is limited data for CSF1R expression in neurons after injury[37], we considered whether PLX3397 or PLX5622 might affect neurons or their expression of tau protein in Tg2541 mice. Acute, chronic, and terminal CSF1R inhibition did not significantly reduce levels of neuronal nuclei (NeuN) detected by IHC, or total tau detected by ELISA (Supplementary Fig. 8). Therefore, CSF1R inhibitors do not directly affect measures of neuronal viability or tau expression, consistent with a prior report using PLX3397 in cultured primary neurons[17]. Together, these data confirm that drug effects on biological and functional end points are due to inhibition of CSF1R in CNS microglia.

## CSF1R inhibition extends survival in female Tg2541 mice

We next focused on the terminal treatment paradigm with PLX3397 to evaluate the long-term effects of CSF1R inhibition on lifespan and behavior. Tg2541 mice develop paraparesis from 5–6 months of age which makes feeding difficult, resulting in a loss of body weight and thus a greatly reduced lifespan compared to wild type mice. We found that terminal PLX treatment significantly extended the median survival of female Tg2541 mice [16.5 days longer median survival; $P = 0.0004$], but not male Tg2541 mice [4.0 days longer median survival; $P = 0.7473$], compared to vehicle treatment (Fig. 2a, b). The extended survival in PLX-treated female mice was preceded by significantly reduced weight loss, which was not observed in male mice (Fig. 2c). As such, body weight at 180 days of age, irrespective of treatment, was predictive of lifespan in female mice and not in male mice, with less weight loss being correlated with longer survival (Fig. 2d). Lower forebrain tau-prion levels were also correlated with longer survival in female mice but not in

male mice (Fig. 2e), suggesting that Tg2541 mice have a sex-specific physiological response to tauopathy.

To confirm the effect of PLX treatment on survival in a different experimental paradigm, we used a midbrain inoculation model. Since Tg2541 mice spontaneously develop substantial tau pathology in the midbrain[27], we predicted that K18 inoculation in the midbrain would accelerate and synchronize the disease course, which would be ideal for studying mouse survival. Indeed, female Tg2541 mice inoculated with K18 tau fibrils died significantly earlier than mice inoculated with diluent, though no difference was observed in male mice (Fig. 2f, g). Consistent with our prior result, PLX treatment significantly extended the median survival of female mice inoculated with K18 tau fibrils [29.5 days longer median survival; $P = 0.0095$], but not male mice [19.5 days shorter median survival; $P = 0.1205$], compared to vehicle treatment (Fig. 2h). Taken together, these data indicate that CSF1R inhibition robustly extends the lifespan of female Tg2541 mice, even during an accelerated disease course.

## CSF1R inhibition reduces aberrant behavioral phenotypes in Tg2541 mice

To examine the relationship between drug exposure and markers of disease progression more closely, we collected blood plasma at monthly intervals from mice receiving terminal treatment with PLX3397 or vehicle (Fig. 3a). Consistent with previous reports[17], male Tg2541 mice had higher plasma (25.3%; $P < 0.0001$) and brain (44.9%; $P = 0.0250$) concentrations of PLX than did female mice (Fig. 3b and Supplementary Fig. 9a–c); we also observed this difference in wild type mice (Supplementary Fig. 9d, e). Male and female mice had *ad libitum* access to food, and had similar rates of food consumption relative to body weight, independent of whether it contained PLX or vehicle (Supplementary Fig. 10a, b). However, female mice were consistently more active than male mice (Supplementary Fig. 10c, d). Thus, the reduced PLX exposure in female mice is likely due to a higher metabolic and drug clearance rate compared with male mice. In line with the hypothesis that drug exposure was excessive in male mice, we observed a trend towards reduced body weight in male wild type mice receiving terminal PLX treatment, but not in female wild type mice (Supplementary Fig. 9f). In spite of this sex-specific difference in PLX exposure, we found that higher plasma concentrations of PLX were correlated with greater microglial depletion in both forebrain and hindbrain regions, independent of sex (Fig. 3c). Furthermore, higher PLX exposure was correlated with reduced tau-prion levels in the forebrain regions of both male and female mice (Fig. 3d). Together, these data indicate that PLX has dose-dependent on-target effects in both male and female mice.

Previous studies have demonstrated a common hyperactive phenotype in the early stages of tauopathy in transgenic rodent models[38,39]. While the precise mechanism that leads to this deficit is unclear, this phenotype is causally linked with tau aggregate burden[40]. Based on the reduction of tau deposition we observed with PLX treatment, we sought to also examine its effect on this hyperactive phenotype. Using an automated home-cage monitoring system, we longitudinally tracked the activity levels of Tg2541 mice at different ages, measuring their amounts of rearing, locomotion, and wheel running. We confirmed the previous reports, finding that at early ages the Tg2541 mice displayed a hyperactive phenotype relative to wild type mice (90–150 days old in females, 90–120 days old in males), while at later ages their activity was significantly reduced (Fig. 3e), likely due to the accumulation of pathogenic tau in brain regions associated with motor function. PLX treatment led to a consistent reduction in Tg2541 mouse hyperactivity, but did not change their hypoactivity at later ages (Fig. 3e), indicating the activity reduction is not due to a general weakening effect. Detailed examination of the individual activity measurements revealed that

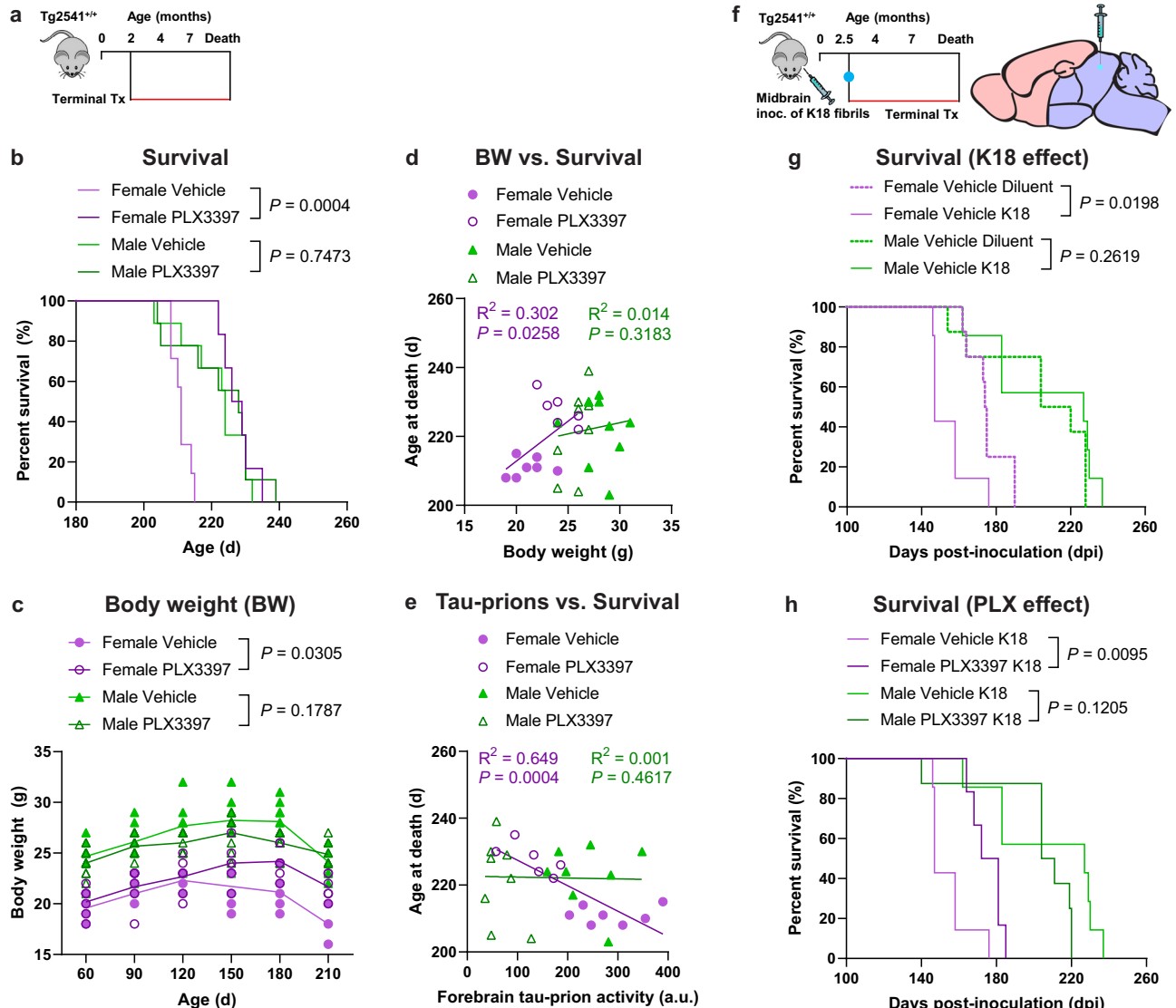

**Fig. 2 | CSF1R inhibition extends survival of female Tg2541 mice. a** Schematic of terminal PLX3397 treatment (275 mg/kg oral) of Tg2541 mice from 2 mo of age until death. **b** Kaplan–Meier plot showing percent survival of female or male Tg2541 mice treated with vehicle or PLX3397. $n = 7$ mice for Female Vehicle; $n = 6$ mice for Female PLX3397; $n = 9$ mice for Male Vehicle; $n = 9$ mice for Male PLX3397. **c** Body weights of female or male Tg2541 mice treated with vehicle or PLX3397. Differences in weight between vehicle and PLX3397 treatment in female or male mice were evaluated by mixed-effects analysis (Restricted maximum likelihood). Each symbol represents an individual mouse and lines indicate group means. **d, e** Correlation plots for body weight at 180 d of age (**d**) or forebrain tau-prion activity at death (**e**) and survival for female or male Tg2541 mice treated with vehicle or PLX3397. Each symbol represents an individual mouse and linear regression lines are shown for female or male mice, with vehicle- and PLX3397-treated mice combined. Pearson's correlation analysis was performed and the results are shown. **f** Schematic of

terminal PLX3397 treatment of Tg2541 mice from 2.5 mo of age until death, following inoculation of K18 tau fibrils into the midbrain (hindbrain region) at 2.5 mo of age. **g** Kaplan–Meier plot showing percent survival of female or male Tg2541 mice inoculated with K18 tau fibrils or diluent. $n = 8$ mice for Female Vehicle Diluent; $n = 7$ mice for Female Vehicle K18; $n = 8$ mice for Male Vehicle Diluent; $n = 7$ mice for Male Vehicle K18. Differences in survival between diluent and K18 inoculation in male or female mice treated with vehicle were evaluated by Log-rank (Mantel-Cox) test. **h** Kaplan–Meier plot showing percent survival of female or male Tg2541 mice inoculated with K18 tau fibrils and then receiving terminal treatment of vehicle or PLX3397 (275 mg/kg oral). $n = 7$ mice for Female Vehicle K18; $n = 6$ mice for Female PLX3397 K18; $n = 7$ mice for Male Vehicle K18; $n = 8$ mice for Male PLX3397 K18. In (**b**, **g**, and **h**), differences in survival between treatment groups were evaluated by Log-rank (Mantel-Cox) test. Source data are provided as a Source Data file.

PLX treatment normalized the amounts of wheel running (Fig. 3f) and active time (Fig. 3g). These data indicate that PLX treatment corrects the aberrant behavior of Tg2541 mice towards that of wild type mice.

**Sex-dependent effects of CSF1R inhibition on a biomarker of CNS injury**

To further interrogate CNS damage caused by tauopathy, or potentially caused by the observed sex-dependent PLX exposure, we also evaluated the plasma levels of neurofilament light chain (NfL). NfL is a

validated blood-based biomarker of neuronal injury[41] which correlates with disease progression and tau burden in human tauopathy[42,43]. Female PLX-treated mice had reduced plasma levels of NfL compared to vehicle-treated mice (Fig. 4a, b), consistent with reduced CNS injury due to tauopathy. Conversely, plasma NfL levels were increased in male mice following PLX treatment (Fig. 4c), suggestive of PLX-induced toxicity. Consistent with these findings, PLX treatment resulted in significantly increased plasma NfL levels in male mice that received midbrain inoculation of K18 tau fibrils, but not female mice (Fig. 4d–f). We found no correlation between plasma NfL level and

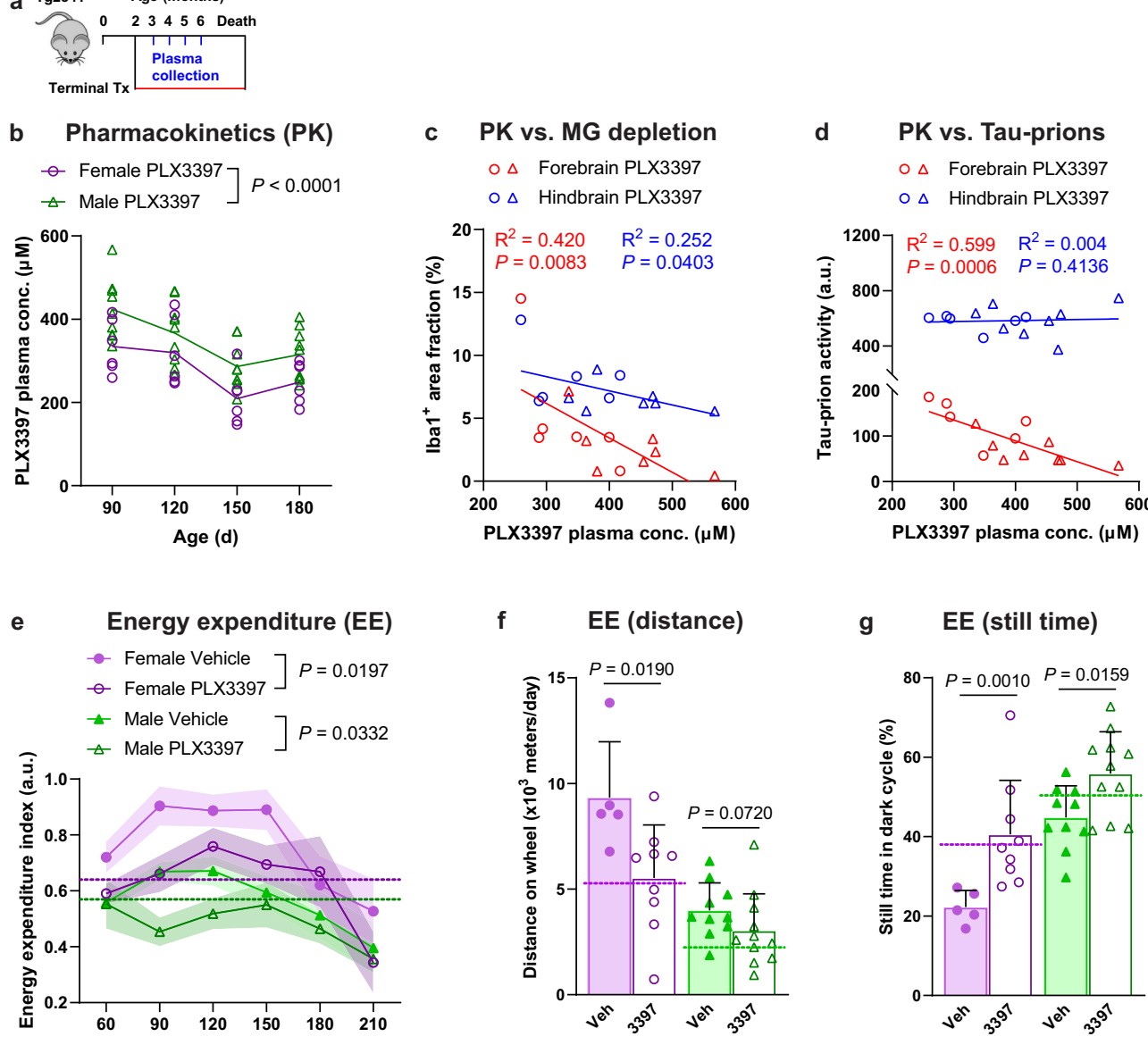

**Fig. 3 | PLX3397 has sex-dependent pharmacokinetics and reduces hyper-activity of Tg2541 mice. a** Schematic of terminal PLX3397 treatment (275 mg/kg oral) of Tg2541 mice from 2 mo of age until death and blood plasma collected at 3, 4, 5, and 6 mo of age. **b** Plasma concentration of PLX3397 in female or male Tg2541 mice. Each symbol represents an individual mouse and the lines indicate group means. The difference between female and male mice was assessed by two-way repeated measures ANOVA. **c, d** Correlation plots for plasma concentration of PLX3397 at 90 d age and Iba1 area fraction by IHC (**c**) or tau-prion activity (**d**) in the forebrains or hindbrains of Tg2541 mice at death. Female mice are shown as open circles and male mice shown as open triangles. Linear regression was performed with female and male mice combined and best-fit lines are shown. Pearson's correlation analysis was performed and the results are shown. **e** Longitudinal energy expenditure indices (see "Methods") of female or male Tg2541 mice treated with vehicle or PLX3397. Symbols represent the group means and shaded regions

indicate the s.d. of the mean. Group sizes are the same as shown in (**f** and **g**). Differences between vehicle and PLX3397 treatment in female or male mice were assessed by three-way repeated measures ANOVA. **f, g** Average distance traveled on the running wheel (**f**) or still time during the dark cycle (**g**) in female or male Tg2541 mice treated with vehicle or PLX3397, and measured between 90-d-old and 150-d-old. Mann–Whitney test was used for each comparison. Each symbol represents an individual mouse and the dashed lines indicate the means of the same measurements in 90-d-old wild type mice. Error bars indicate the s.d. of the mean. Differences between vehicle and PLX3397 treatment in female or male mice were assessed by Mann–Whitney test. Data are presented as mean ± S.D. In (**e**–**g**), $n = 5$ mice for female vehicle group, 9 mice for female PLX group, 10 mice for male vehicle group and 11 mice for male PLX group. Source data are provided as a Source Data file.

survival in female mice, but in male mice, higher plasma NfL levels were clearly correlated with reduced survival in both PLX and vehicle treatment groups (Fig. 4g, h). Furthermore, the brain concentrations of NfL in both the forebrain and hindbrain at death were positively correlated with PLX3397 concentration in male, but not female, Tg2541 mice (Fig. 4i, j). Interestingly, intermittent treatment, resulting in a 50% lower total dosage of PLX, produced a significant decrease in

plasma NfL levels in male mice, but had no effect in female mice (Fig. 4k–m). Taken together, these data suggest that in female Tg2541 mice, tauopathy drives CNS injury and its reduction by PLX effectively masks any effect of PLX toxicity, whereas in male mice, excessive exposure causes CNS injury that supersedes any benefit of PLX. Consistent with this premise, we found that PLX treatment increased plasma NfL levels in male wild type mice, as expected, but also in

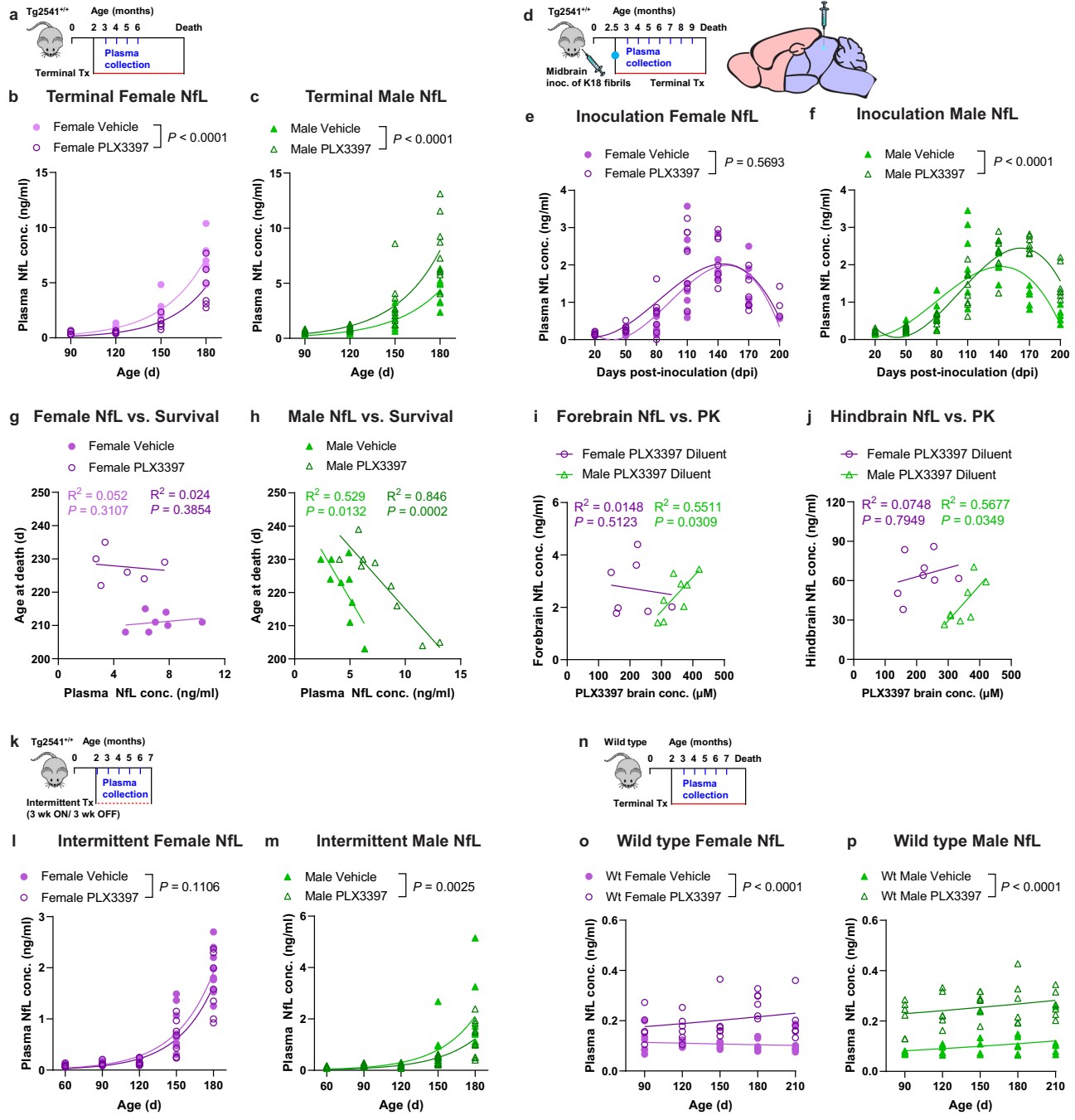

**Fig. 4 | Plasma NfL levels are reduced by PLX3397 in female Tg2541 mice and increased in male mice. a** Schematic of terminal PLX3397 treatment (275 mg/kg oral) of Tg2541 mice from 2 mo of age until death and blood plasma collected at 3, 4, 5, and 6 mo of age. **b, c** Plasma concentration of NfL in female (**b**) or male (**c**) Tg2541 mice treated with vehicle or PLX3397. Groups were compared by nonlinear regression. **d** Schematic of terminal PLX3397 treatment of Tg2541 mice from 2.5 mo of age until death, following inoculation of K18 tau fibrils into the midbrain (hindbrain region) at 2.5 mo of age, and blood plasma collected monthly thereafter until death. **e, f** Plasma concentration of NfL in female (**e**) or male (**f**) Tg2541 mice inoculated with K18 tau fibrils followed by terminal treatment with vehicle or PLX3397, plotted over days post-inoculation (dpi). **g, h** Correlation plots for plasma NfL concentration and survival in female (**g**) or male (**h**) Tg2541 mice. Each symbol represents an individual mouse. **i, j** Correlation plots for brain concentration of PLX3397 and NfL concentration in the (**i**) forebrains or (**j**) hindbrains of Tg2541 mice receiving midbrain inoculation with diluent. Each symbol represents the forebrain or hindbrain of an individual mouse. In (**g–j**), linear regression and Pearson's correlation analyses were performed and the results are shown. **k** Schematic of intermittent PLX treatment of Tg2541 mice from 2–7 mo of age, with three weeks on treatment followed by three weeks off of treatment, and blood plasma collected at 2, 3, 4, 5, and 6 mo of age. **l, m** Plasma concentration of NfL in female (**l**) or male (**m**) Tg2541 mice receiving intermittent treatment with vehicle or PLX3397. Groups were compared by nonlinear regression. **n** Schematic of terminal PLX treatment of C57BL/6 J wild type mice (Wt) from 2 mo of age until death and blood plasma collected at 3, 4, 5, 6, and 7 mo of age. **o, p** Plasma concentration of NfL in female (**o**) or male (**p**) Wt mice treated with vehicle or PLX3397. Groups were compared by nonlinear regression. In (**b, c, e, f, l, m, o**, and **p**), the differences between vehicle or PLX3397 treatment were evaluated by nonlinear regression using exponential growth models (**b, c, l, m, o**, and **p**) or third-order polynomial models (**e** and **f**) and the best-fit lines and statistical results are shown. Each symbol represents an individual mouse. Source data are provided as a Source Data file.

female wild type mice (Fig. 4n–p), albeit at substantially lower levels than tau-induced NfL in Tg2541 mice.

## Peripheral CSF1R inhibition does not cause therapeutic or adverse effects in the CNS

CSF1R inhibitors have known adverse effects in the periphery including anemia, leukopenia, and hepatotoxicity, which have been observed in human clinical trials[44]. To determine if peripheral PLX toxicity was causing NfL release or reducing drug efficacy, we evaluated mice receiving chronic treatment with PLX73086, the non-brain penetrant CSF1R inhibitor analog of PLX3397 and PLX5622. We observed no effect of PLX73086 treatment on body weight or plasma NfL levels in either male or female Tg2541 mice (Fig. 5a–d), indicating that peripheral CSF1R inhibition does not impact tauopathy-driven phenotypes. Furthermore, we observed no effect of PLX73086 treatment on body weight or plasma NfL levels in wild type mice (Fig. 5e–h), demonstrating that the drug-induced toxicity indicated by elevated NfL is dependent on the drug entering the brain. Histopathological evaluation of liver sections of PLX3397-dosed Tg2541 mice stained with hematoxylin and eosin (H&E), Masson's trichrome, or Picosirius red did not reveal any overt signs of liver injury or fibrosis (Fig. 5j), nor did quantification of the Picosirius red-stained liver images (Fig. 5k). Alkaline phosphatase (ALP) in the plasma, a different indicator of liver damage, was elevated in Tg2541 mice following PLX3397 treatment; however, the ALP levels were elevated in both male and female mice (Fig. 5l), indicating that the sex-specific toxicity of PLX3397 in male mice likely occurs in the brain rather than in the periphery. Taken together, these data further confirm that the drug exposure of brain-penetrant CSF1R inhibitors was appropriate for female Tg2541 mice to reduce tauopathy and significantly extend survival; however, the drug exposure was too high for male Tg2541 mice, and the resulting neurotoxicity outweighed its therapeutic benefit.

## CSF1R inhibition shifts gene expression patterns in Tg2541 mice towards wild type

To better characterize global molecular changes in the CNS due to CSF1R inhibition, we used the Nanostring platform to analyze a curated panel of gene transcripts related to neuroinflammation, myeloid cell function and neuropathology in bulk brain tissue following chronic treatment of Tg2541 mice with PLX5622. The chronic treatment group was selected for this analysis because the collection time point was synchronized (unlike in the terminal treatment group) and seven months of age is close to the average lifespan of Tg2541 mice. We measured mRNA transcripts of 1841 genes, many of them shown to be regulated by tau or Aβ pathology in previous genome-wide gene expression studies[12,13]. To validate the Nanostring approach, we identified 53 genes with a broad range of expression level changes and measured their mRNA transcripts by quantitative reverse-transcription PCR (RT-qPCR) in the same samples used for sequencing. The RT-qPCR results generally matched the trends shown in the Nanostring data (Supplementary Fig. 11), supporting the validity of our transcriptomic data.

Since microglia are directly impacted by PLX treatment, we first excluded the microglial-specific genes (see "Methods") and examined the general trend of expression patterns among different treatment groups. Pearson's correlation matrix showed high similarity among wild type brains with or without PLX treatment (Fig. 6a), indicating that the gene expression pattern we measured is not affected by the treatment itself. In contrast, PLX treatment in Tg2541 mice caused a distinct shift in the gene expression pattern away from the vehicle-treated group. The gene expression patterns of PLX-treated Tg2541 mice showed a greater correlation with wild type mice than with vehicle-treated Tg2541 mice (Fig. 6a dashed boxes and Fig. 6b). To further quantify this shift, we performed partial-least squares (PLS)

regression analysis using the gene expression data from vehicle-treated Tg2541 and wild type mice (Fig. 6c filled circles), and projected the data from PLX-treated mice onto the PLS dimensions (Fig. 6c empty circles). This allowed us to represent the transgene-specific gene expression pattern in a relatively low-dimensional space, and to quantify the changes associated with treatment by calculating the population vector distances and angles in this space. We found that PLX treatment significantly normalized gene expression patterns in Tg2541 mice towards those of wild type mice (Fig. 6d, e, only two out of five dimensions are shown, covering >95% of the total variance). The normalization in gene expression was further confirmed by similar trends in neuron-specific genes (Supplementary Fig. 12). Importantly, PLX treatment in wild type mice showed negligible changes in the gene expression patterns. These results indicate that PLX treatment specifically suppresses the abnormal transcriptome associated with transgene overexpression, consistent with its effects ameliorating pathogenic tau deposition.

## Evidence for excitotoxicity with increased drug exposure

As described above, although we observed consistent reduction in the levels of pathogenic tau in the brains of both male and female Tg2541 mice with PLX treatment, only female mice benefited from extended survival, functional rescue, and reduced NfL levels (Figs. 1–4). We hypothesized that excessive PLX dosing may underlie this sex-specific effect, as male mice consistently had higher drug exposure in the plasma and CNS (Fig. 3 and Supplementary Fig. 9), and also appeared to benefit from a lower PLX dosage in the intermittent dosing paradigm (Fig. 4m). First, we ruled out the possibility that sex-dependent functional benefits were caused by differential expression of the drug target, CSF1R, or its ligand, CSF1 (Supplementary Fig. 13). Then, in our transcriptomic analysis we identified individual genes whose expression was associated with brain PLX5622 concentration or with sex (Fig. 6f). Using a PLS regression of all non-microglia genes to brain PLX concentration and sex for each sample, we calculated the variable importance score along each of these dimensions. Unexpectedly, many immediate early genes (IEGs) showed high importance scores (Fig. 6g), suggesting that IEGs might be a module that is altered by CNS drug exposure. To further test this possibility, we examined all IEGs (56 genes overlapped in our dataset) and found that their expression patterns fit closely with the brain PLX concentration (Fig. 6h). Importantly, when we excluded the IEGs and examined the PLX-induced transcriptome shift along the wild type-to-Tg2541 dimension, the correlation of gene expression changes and brain PLX concentration was no longer significant (Fig. 6i), indicating that the IEGs contribute substantially to the PLX treatment effects. Notably, relative to wild type vehicle-treated mice, only male PLX5622-treated Tg2541 mice had significantly upregulated expression of IEGs (Fig. 6j), which we also validated using qPCR for the five most highly expressed IEGs (Supplementary Fig. 14a–c), as well as by in situ labeling of *Fos* mRNA (Fig. 6k, l). Furthermore, we observed the same drug-dependent IEG upregulation in male mice in a completely independent study using PLX3397 (Supplementary Fig. 14d–f), thereby confirming this robust sex-specific effect of CSF1R inhibition. As increased IEG expression can be indicative of neuronal hyperactivity, these data provide a plausible mechanism by which excessive PLX dosing may have led to excitotoxicity, thereby masking its therapeutic effect in male mice.

To interrogate this mechanism directly, we performed hydrophilic interaction liquid chromatography tandem mass spectrometry (HILIC-MS/MS)-based metabolomics to measure the levels of relevant neurotransmitters in mouse forebrain lysates. Male PLX3397-treated Tg2541 mice, but not PLX-treated wild type or female Tg2541 mice, had increased brain levels of the excitatory neurotransmitter glutamate (Fig. 7a), the most common cause of neuronal excitotoxicity[45]. Unexpectedly, PLX-treated male Tg2541 mice also had increased levels of the inhibitory neurotransmitter γ-aminobutyric acid (GABA) (Fig. 7b).

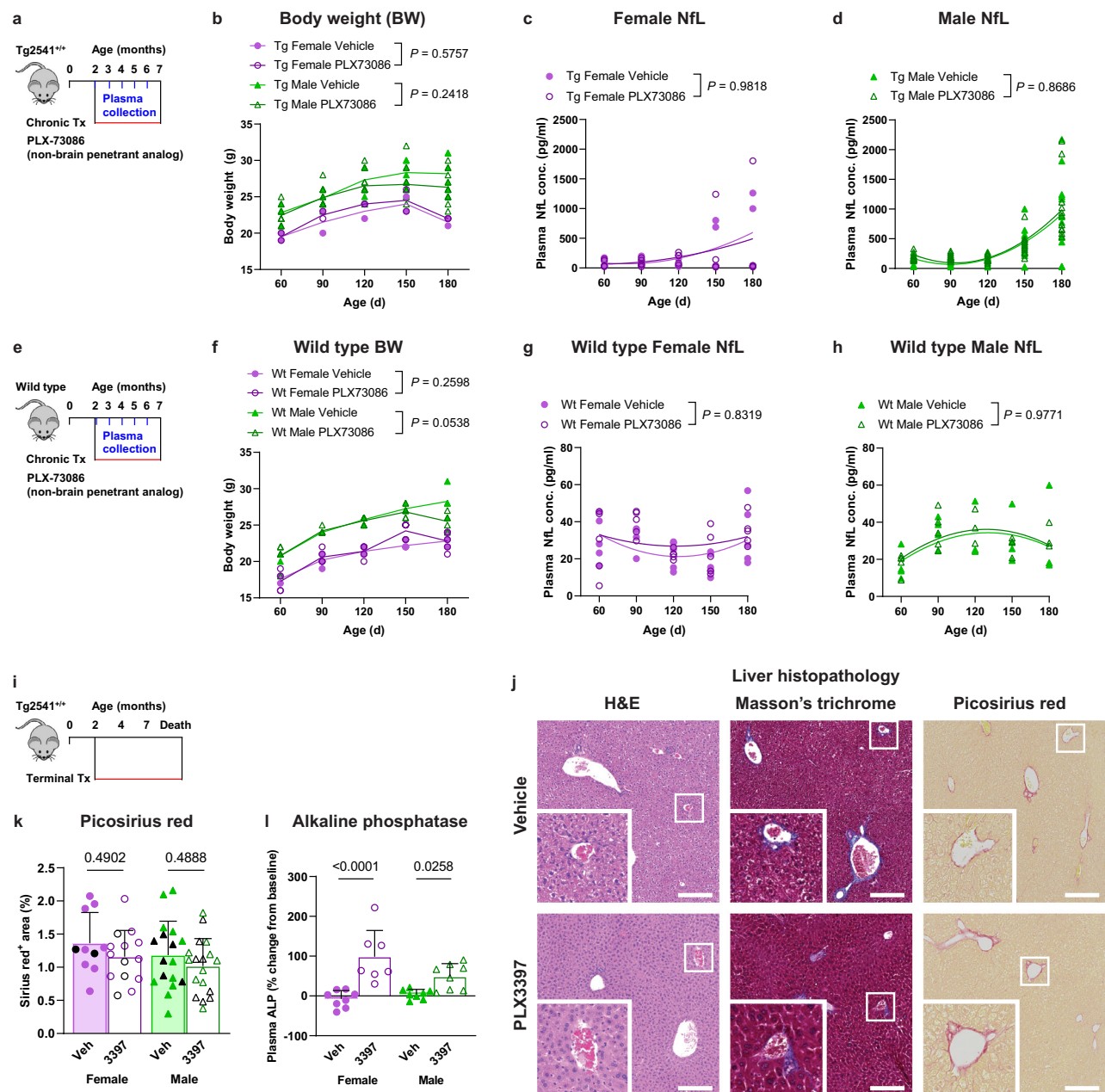

**Fig. 5 | Sex-dependent effects of CSF1R are not caused by peripheral toxicity.**
**a** Schematic of chronic treatment of Tg2541 mice from 2–7 mo of age with PLX73086 (200 mg/kg oral), a non-brain penetrant analog of PLX3397 and PLX5622. **b** Body weights of female or male T2541 mice treated with PLX73086. **c, d** Plasma concentration of NfL in female (**c**) or male (**d**) Tg2541 mice treated with vehicle or PLX73086. **e** Schematic of chronic treatment of wild type (Wt) mice from 2–7 mo of age with PLX73086 (200 mg/kg oral). **f** Body weights of female or male Wt mice treated with PLX73086. In (**b** and **f**), differences in weight between vehicle and PLX73086 treatment in male or female mice were evaluated by two-way repeated measures ANOVA and P values are shown. Each symbol represents an individual mouse and lines indicate group means. **g, h** Plasma concentration of NfL in female (**g**) or male (**h**) Wt mice treated with vehicle or PLX73086. In (**c, d, g,** and **h**), the differences between vehicle or PLX3397 treatment were evaluated by non-linear regression using quadratic models. Each symbol represents an individual mouse and the best-fit lines and statistical results are shown. **i** Schematic of terminal treatment of Tg2541 mice from 2 mo of age until death with PLX3397. **j** Representative histopathology images of liver sections of Tg2541 mice receiving terminal treatment with vehicle or PLX3397, stained with hematoxylin and eosin (H&E), Masson's trichrome, or Picosirius red. High magnification insets are shown of the regions outlined with a white box. Scale bars, 200 μm. H&E and Masson's

trichrome images are representative of eight vehicle-treated mice (two females and six males) and 12 PLX3397-treated mice (four females and eight males) analyzed. Sirius red images shown are representative of images collected and analyzed for all mice shown in panel (**k**). **k** Sirius red-stained liver sections of Tg2541 mice that received acute or terminal treatment with vehicle or PLX3397 were quantified for percent positive area. The acute and terminal treatment groups were combined for the analysis due to a limited sample size of terminal treatment groups and because the group means were similar for the two treatment paradigms. Mice receiving terminal treatment are shown as black symbols and mice receiving acute treatment are shown as purple or green symbols. n = 10 female vehicle, 14 female PLX3397, 17 male vehicle, and 16 male PLX3397-treated mice. **l** Alkaline phosphatase (ALP) levels were measured at eight months of age in the plasma of Tg2541 mice receiving midbrain inoculation of diluent at 2.5 mo of age and then treated with vehicle or PLX3397 until death. The data are presented as a percent change from ALP levels at two months of age (prior to treatment) in the same mice. n = 8 female vehicle, 7 female PLX3397, 8 male vehicle, and 8 male PLX3397-treated mice. In (**k** and **l**), the differences between vehicle and PLX3397 treatment were evaluated by ANOVA with Holm-Šidák post hoc analysis and P values are shown. Each symbol represents an individual mouse and error bars indicate the s.d. of the mean. Source data are provided as a Source Data file.

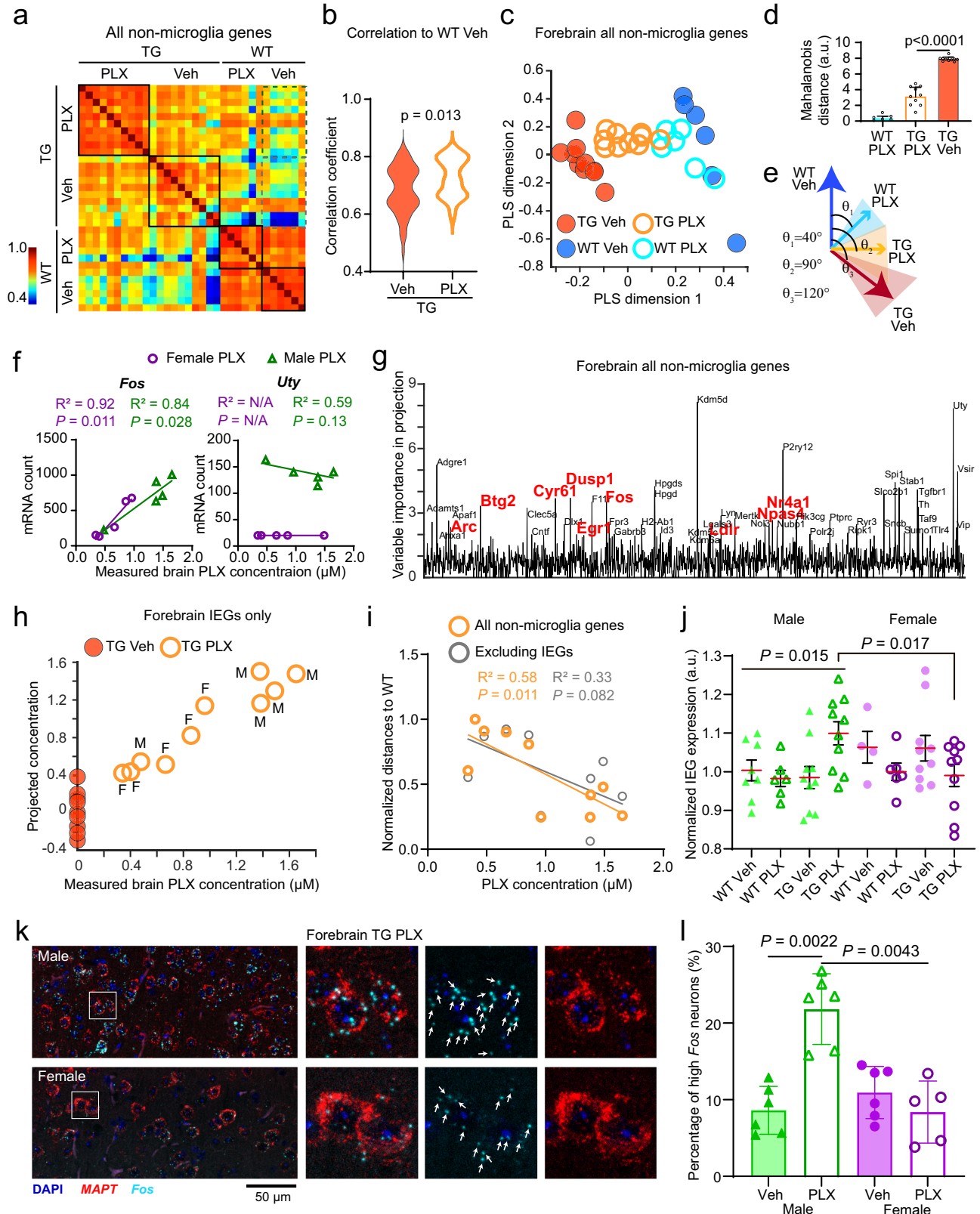

However, these PLX-induced alterations were not due to general neurotransmitter dysregulation because we did not observe altered levels of acetylcholine, glycine, or histamine in the brains of male Tg2541 mice (Supplementary Fig. 15). We also measured dopamine, serotonin, and norepinephrine; however, their levels were below the limit of detection. These data provide evidence of PLX-induced glutamate-driven excitotoxicity in the brains of male Tg2541 mice which

may, in part, underlie the sex-specific benefit of CSF1R inhibition on mouse survival.

## CSF1R inhibition ameliorates pathological activation of astrocytes

As suggested by the reduction of tau deposition, we hypothesized that astrocyte-driven neuroinflammation would also be reduced by PLX

**Fig. 6 | PLX treatment normalizes gene expression profile, but dose-dependent excitotoxicity occurs with higher drug exposure. a** Pearson's correlations among individual mice for non-microglia gene expression patterns (1599 genes for each mouse). Mice from the same treatment group (black boxes) show a high degree of correlation. Dashed boxes show correlations of Tg2541 mice with wild type vehicle-treated mice. **b** Distributions of correlation coefficients in the two groups in dashed boxes in panel **a**. Two-sided *t*-test was used for comparison. **c** Partial least square (PLS) regression scores and projections of non-microglia gene expression patterns in different groups (1599 genes for each mouse). The first two dimensions are shown covering >95% of the total variance. Vehicle-treated mice (filled dots) were used for regression (five dimensions covering 99.99% of the total variance) and PLX5622-treated (1200 mg/kg oral) mice (empty circles) were projected onto the regression dimensions. **d** Mahalanobis population vector distances of gene expression patterns in PLS dimensions relative to the wild type vehicle group. Data are presented as mean ± S.D. Mann−Whitney test was used for statistical comparison. **e** Population vector angles of gene expression patterns in PLS dimensions from different treatment groups. **f** Example plots of measured brain PLX concentration against mRNA counts for a PLX-modulated gene (*Fos*) and a sex-modulated gene (*Uty*). Linear regression and Pearson's correlation analysis were performed and the results are shown. N/A indicates that there was no detectable expression of the *Uty* gene in female mice. See Supplementary Data File 2 for statistical comparisons between male and female PLX-treated mice of all genes. **g** Variable importance in projection (VIP) scores in a PLS regression using non-

microglial gene expression patterns to brain PLX concentration and sex. Genes with VIP scores above 2 are labeled. Red fonts indicate a subset that belongs to immediate early genes (IEGs). **h** Scatter plot of measured vs. projected brain PLX concentration, which was based on a PLS regression with IEG expression. **i** Correlation of brain PLX concentration and the population vector distances between PLX-treated individual mouse transcriptome pattern to WT, with and without the IEGs (1599 and 1543 genes for each mouse, respectively). Linear regressions were calculated for each group. **j** Quantification of normalized expression levels from all IEGs (56 genes for each mouse) in the forebrain. Data are presented as mean ± S.E.M. One-way ANOVA was used to compare different groups in male mice and two-sided *t*-test was used to compare between male TG PLX group to female TG PLX group. **k** Representative confocal images of in situ labeling of *MAPT* and *Fos* mRNA (RNAscope) in the forebrains of male and female Tg2541 mice treated with PLX3397 from 2 mo of age until death. Zoomed regions of interest are shown to the right. Arrows indicate labeled *Fos* puncta. **l** Quantification of *Fos* mRNA levels measured in RNAscope images shown in (**k**). Data are presented as mean ± S.D. Mann−Whitney tests were used for comparing between groups. In (**c**, **f**, **h**−**j**, and **l**), each symbol represents an individual mouse. In (**c**−**e**, **h**) and (**i**), *n* = 10 mice for TG groups, and *n* = 6 mice for WT groups. In (**j**), *n* = 8, 6, 10, 10, 4, 6, 10, 10 mice for each group, in the order on the *x*-axis. In (**l**), *n* = 6 mice for each group except for *n* = 5 mice for PLX female group. Source data are provided as a Source Data file.

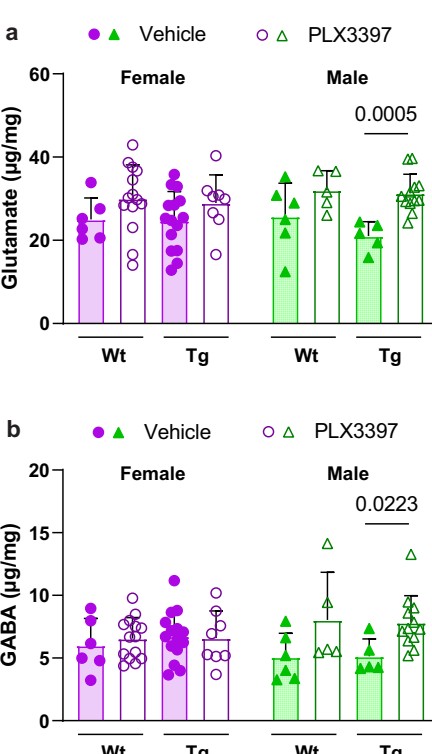

**Fig. 7 | PLX dysregulates neurotransmitter levels in the brains of male Tg2541 mice. a** Glutamate levels in forebrain lysates of wild type (Wt) or Tg2541 (Tg) mice, dosed for 6–12 weeks with PLX3397 (275 mg/kg oral) or vehicle beginning at 4 months of age, were measured by hydrophilic interaction liquid chromatography tandem mass spectrometry (HILIC-MS/MS). **b** γ-Aminobutyric acid (GABA) levels in forebrain lysates of Wt or Tg mice, dosed for 6–12 weeks with PLX3397 (275 mg/kg oral) or vehicle beginning at 4 months of age, were measured HILIC-MS/MS. In (**a** and **b**), Mann−Whitney tests were used for comparing vehicle and PLX3397 treatment groups within mouse genotype and sex, and the *P* values of statistically significant differences (*P* < 0.05) are shown. Each symbol represents an individual mouse and error bars indicate the s.d. of the mean. *n* = 6 Wt female vehicle, 14 Wt female PLX3397, 6 Wt male vehicle, 5 Wt male PLX3397, 15 Tg female vehicle, 8 Tg female PLX3397, 5 Tg male vehicle, and 12 Tg male PLX3397-treated mice. Source data are provided as a Source Data file. Raw data files from all metabolomics analyses are provided as Supplementary Data File 3.

treatment. Therefore, we examined transcriptome shifts in astrocyte-specific genes upon PLX treatment. Similar to the neuronal-specific genes, we observed a normalization of astrocyte-specific gene expression patterns towards wild type in both forebrain and hindbrain regions (Fig. 8a, b). Using previously described astrocytic gene signatures of disease[46], we found that PLX treatment led to a dose-dependent reduction in the expression of the A1 astrocytic gene cluster associated with neurotoxic effects (Fig. 8c, d). In addition, we measured astrogliosis over time using longitudinal bioluminescence imaging (BLI) methods based on a previously established transgenic reporter system of glial fibrillary acidic protein (GFAP)-driven luciferase[47], which we validated by IHC and mRNA analyses (Supplementary Fig. 16). To perform reliable BLI in Tg2541 mice, we intercrossed each transgenic line to an albino background and refined the method, using a synthetic luciferin substrate to increase signal from deep hindbrain regions (see "Methods" and Supplementary Fig. 17). This technique allowed us to non-invasively measure astrogliosis in live mice longitudinally over the course of PLX treatment. In vehicle-treated Tg2541 mice, the BLI signal gradually increased with age (Supplementary Fig. 18), in accordance with the accumulation of tau pathology and gliosis reported in Tg2541 mice[23,48]. Consistent with our hypothesis and with measurement of GFAP using other methods (Supplementary Fig. 16), CSF1R inhibition suppressed the BLI signal in both the forebrain and the hindbrain (Fig. 8e−g). Together, these data suggest that astrocytic inflammation, specifically neurotoxic astrocytes driven by microgliosis, was attenuated by CSF1R inhibition, thus leading to a general neuroprotective effect.

## CSF1R inhibition preferentially eliminates a highly activated microglia subpopulation

Given reported roles of microglia in driving astrocytic inflammation and neurotoxicity in disease, we next interrogated whether resilient microglial phenotypes could be responsible for the sex-specific neurofunctional effects. Thus, we examined the morphological and transcriptional changes in microglia following CSF1R inhibition. In tauopathy, microglia acquire an activated morphology in brain regions where neurons contain tau aggregates, a phenomenon seen in many focal neuropathologies[49]. Interestingly, the elimination of microglia in the Tg2541 mouse brain following PLX3397 treatment was not uniform nor complete, but was the most effective in the vicinity of tau aggregates (Fig. 9a, b). The microglial density near tau-laden neurons was reduced by more than 60%, but in distal regions (>200 microns) the

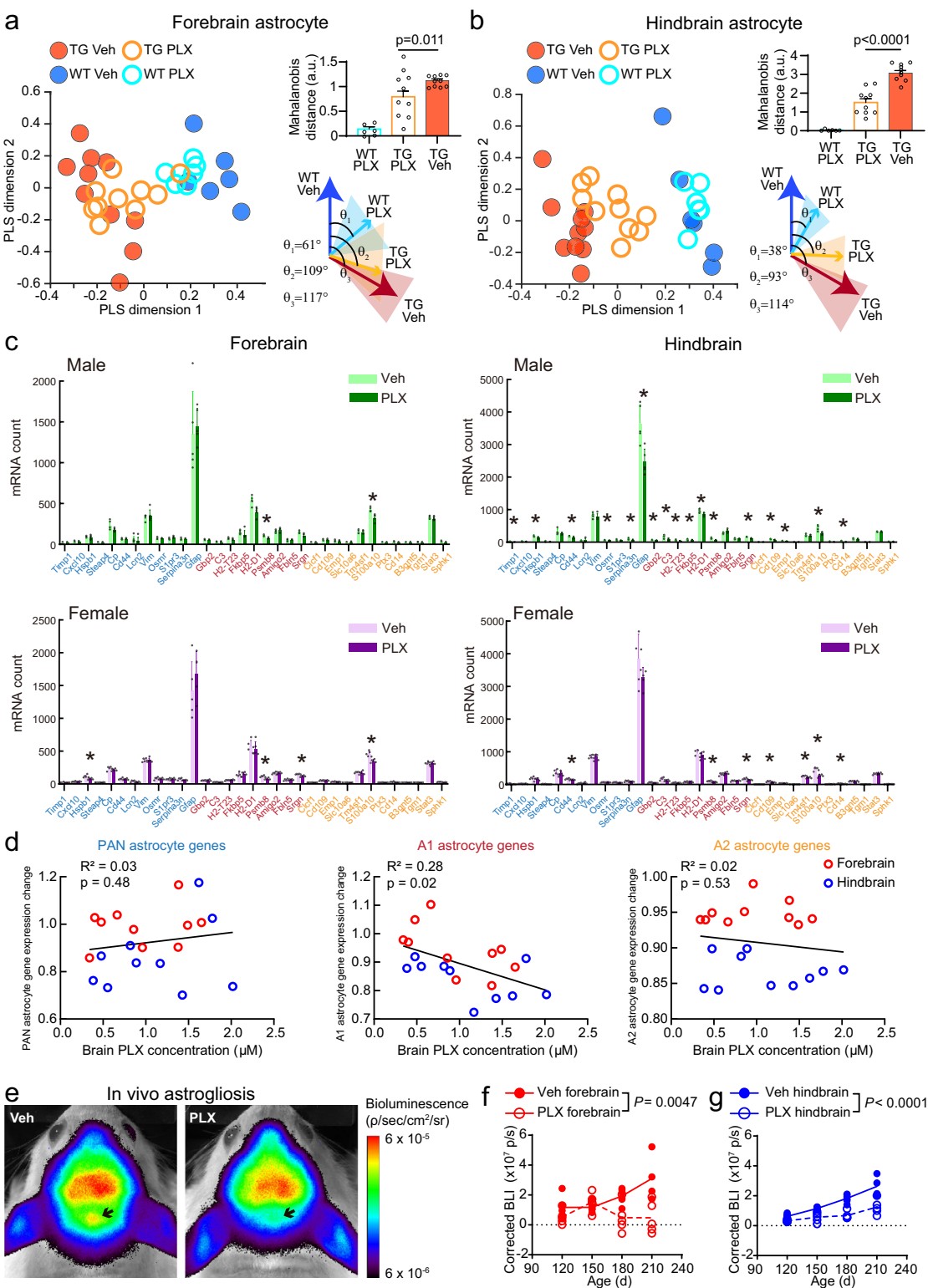

**Fig. 8 | CSF1R inhibition ameliorates tau-induced pathological astrocyte activation. a, b** Analyses similar to Fig. 6c–e using astrocyte-specific genes (47 genes in each mouse) in (**a**) forebrain and (**b**) hindbrain regions. For other cell types and brain regions, see Supplementary Fig. 12. In (**a** and **b**), data are presented as mean ± S.D., Mann–Whitney tests were used for comparison. **c** Quantifications of featured astrocyte genes in different conditions. Data are represented as mean ± S.D. Two-sided *t*-tests were used to compare vehicle and PLX-treated groups for each gene, with 5% false discovery rate correction for multiple comparisons. Asterisks indicate *P* < 0.05. **d** Correlation of brain PLX concentration and different groups of astrocyte gene expression. Based on a prior study[46], A1 genes are associated with neurotoxic astrocytes following lipopolysaccharide exposure and A2 genes are associated with

neuroprotective function in an artery occlusion model. Linear regressions were calculated for each group. **e** Representative images of in vivo bioluminescence imaging of GFAP activity in Tg2541 mice with vehicle or PLX3397 treatment. Arrows indicate hindbrain regions. **f, g** Quantifications of longitudinal measurements of astrogliosis-driven bioluminescence in the (**f**) forebrains or (**g**) hindbrains of Tg2541 mice with vehicle or PLX3397 treatment. Differences in BLI signal between vehicle and PLX treatment were evaluated by mixed-effects analysis (Restricted maximum likelihood). In (**a**, **b**, **d**, **f**, and **g**) each symbol represents an individual mouse. In (**a** and **b**), *n* = 10 mice for TG groups, and *n* = 6 mice for WT groups. In (**c**), *n* = 5 mice for each group. In (**f** and **g**), *n* = 5 mice for PLX3397 group and *n* = 6 mice for Vehicle group. Source data are provided as a Source Data file.

microglial density was not significantly changed (Fig. 9c), indicating that microglia in the vicinity of the tau aggregates may have increased sensitivity towards CSF1R inhibition. We then compared the morphologies of PLX-resistant (residual microglia in PLX-treated mice) and PLX-sensitive microglia (in vehicle-treated mice). Notably, we found that PLX resistance was associated with more abundant and intricate microglial cell processes, close to the levels seen in wild type mice (Fig. 9d–g). The number of microglial process branches ($P = 0.0075$), process lengths ($P = 0.028$), and territory sizes ($P = 0.011$) in the forebrain were also significantly different between male and female PLX-treated mice, with female mice showing more abundant and intricate microglial processes by all three metrics (Fig. 9d–g). These data suggest that microglia associated with tau pathology may be in an "activated" state, with a reduced number of processes. This view is consistent with previous reports that activated microglia adopt a "disease-associated microglial" (DAM) phenotype, with some of the signature genes associated with inflammatory responses that are detrimental to adjacent neural cells[50,51]. Our data suggests that DAM in tauopathy are more vulnerable to PLX treatment, and that surviving microglia might be neuroprotective, particularly in female mice.

In support of this view, transcriptome analyses showed that many microglial-specific genes were upregulated in Tg2541 mice (Fig. 10a), among which the most notable were signature DAM genes such as *Tyrobp*, *Clec7a*, *Trem2* and *CD68*. By correlation analysis among different samples using a generalized Louvain algorithm[52], we found that the microglial-specific genes in our dataset were clustered into three groups (Fig. 10b and Supplementary Fig. 19). The red-cluster genes showed the highest degree of modulation by transgene over-expression, while the blue-cluster genes showed a moderate degree of modulation, and the green-cluster genes showed almost no difference between Tg2541 and wild type mice (Fig. 10b). Transgene-modulated genes clustered into red and blue groups, consistent with a recent finding that tau pathology activates both immune-activation and immune-suppression gene expression modules[10].

We next compared the gene expression patterns in vehicle- and PLX5622-treated brains. We examined all previously reported DAM signature genes[51] and found a partial match with the activation markers in each of our identified gene clusters (Fig. 10c). Regardless of the designation of homeostatic or activation genes reported in previous studies, the red-cluster genes showed a stereotypical pattern of transgene activation and sensitivity to PLX treatment, while green- and blue-cluster genes did not appear to be modulated by these factors. Given that PLX predominantly eliminated microglia in the vicinity of the tau deposits (Fig. 9c), these data suggest that red-cluster genes are preferentially expressed by tau-associated microglia. When we examined PLX-induced expression changes in male and female mice separately, we found that while the expression of red cluster genes were substantially diminished by PLX independent of sex, green cluster genes and some blue cluster genes were markedly increased in treated male mice (Fig. 10d, e). We validated some of the sex-specific microglial genes affected by PLX treatment using real-time PCR as well as by in situ labeling of mRNA (Fig. 10f, g). To better describe the microglial gene expression in the surviving population and account for the cell number reduction, we calculated the gene expression of resilient microglia by normalization to six microglia-specific housekeeping genes. The normalized data showed a trend towards increased gene expression in male mice, yet most of the male-specific PLX-induced genes belonged to the green cluster (Fig. 10e), many of which are known to be pro-inflammatory. Indeed, subsequent Ingenuity Pathway Analysis revealed a higher activation of inflammation-related pathways in PLX-treated male mice compared to female mice (Fig. 10h). These inflammation-related pathways included tau-activated NF-κB[53] and excitotoxicity[54] pathways, but not amyloid-induced inflammasome genes[55] (Supplementary Fig. 20). Pathways related to phagocytosis or microglial growth, on the other hand, did not show a consistent sex-specific change (Fig. 10h). Taken together, our data show morphological and transcriptional changes in microglia associated with tau deposition, consistent with a pattern of pathological activation. CSF1R inhibition appears to preferentially eliminate these microglia in female mice, leaving the brain with a more quiescent and less inflammatory microglial population. Male mice, on the other hand, showed a drug-induced inflammatory microglial phenotype, which might contribute to neuronal excitotoxicity and diminished therapeutic effect.

## Discussion

Our study reveals several important findings from a comprehensive evaluation of CSF1R inhibitors in preclinical models of tauopathy. Importantly, we present evidence that CSF1R inhibition reduces pathology that leads to functional improvements associated with longer lifespan and reduced behavioral deficits in tauopathy mice (Figs. 2 and 3). Overall, our data showing a reduction of pathogenic tau is consistent with prior studies using a different drug scaffold targeting CSF1R (JNJ-527; edicotinib) in Tg2541 mice[18], or using PLX3397 in a different mouse model of tauopathy (TgPS19)[17]. However, in our study, neuroprotection occurred despite incomplete microglia depletion (~60%); upon deeper analysis, we identified distinct microglial-specific gene clusters suggesting subsets of microglia responsive to tauopathy or resilient to CSF1R inhibition (Fig. 10). This finding is in line with the wealth of data demonstrating that microglia exist as unique subsets in different brain regions, sexes, ages, or disease states[2–4]. From this perspective, our data suggest that it may be possible to target specific subsets of activated microglia in tauopathy, without affecting other beneficial microglial populations. Taken together, our data argue that complete microglia ablation is unnecessary for therapeutic benefits of CSF1R inhibition, and furthermore may possibly be detrimental in humans given that microglia are important for brain homeostasis and defending against other insults. Several prior studies have suggested that therapeutic outcomes may only be achieved with complete ablation of microglia, although these studies used different mouse models or dosing regimens[17,19]. Nevertheless, these disparate findings in prior literature are now more interpretable alongside our study, which sheds light on the intricate relationships between CSF1R inhibitor dosing, microglial depletion and therapeutic outcomes.

The precise mechanism of CSF1R inhibitors causing reduced tau pathology is still unclear, but our data indicates that activated microglia are the primary target resulting in reduced numbers of cells producing pro-inflammatory cytokines[7–9] and other disease-associated microglial factors (Fig. 10) that stoke tau pathogenesis in neurons, such as apolipoprotein E[17,56] and complement proteins[57]. In addition, it seems plausible that CSF1R inhibitors may also block microglia-mediated activation of A1 astrocytes[46], which in turn secrete factors that also drive tau pathogenesis; blocking this cellular feed-forward pathway using a different drug targeting microglia led to neuroprotection in a synucleinopathy model of Parkinson's disease[58]. Consistent with this view, PLX-treated mice exhibited a restored astrocyte phenotype, suggesting that therapeutic benefits in our study may also be due, in part, to quelling neurotoxic astrocytes (Fig. 8). Pharmacological CSF1R inhibition has been reported to also deplete PVMs[59]. Although we did not detect a significant reduction in the numbers of cortical blood vessel-associated PVMs in our studies (Supplementary Fig. 7), their roles in tauopathy remain to be determined. While there is cross-talk between peripheral immune cells and microglia[1], we show that a non-brain penetrant analog, PLX73086, did not affect CNS microglia, tau pathology or NfL levels (Fig. 5 and Supplementary Fig. 6), and it is thus unlikely that CSF1R inhibition in the periphery contributes to the phenotypic rescue observed in our study. Lastly, it is possible that chronic PLX treatment caused depletion of some oligodendrocyte progenitor cells (OPCs) in our study, but we expect that PLX did not affect mature oligodendrocytes or myelination[60]. The relationship between OPC biology and tau pathology in neurons is largely

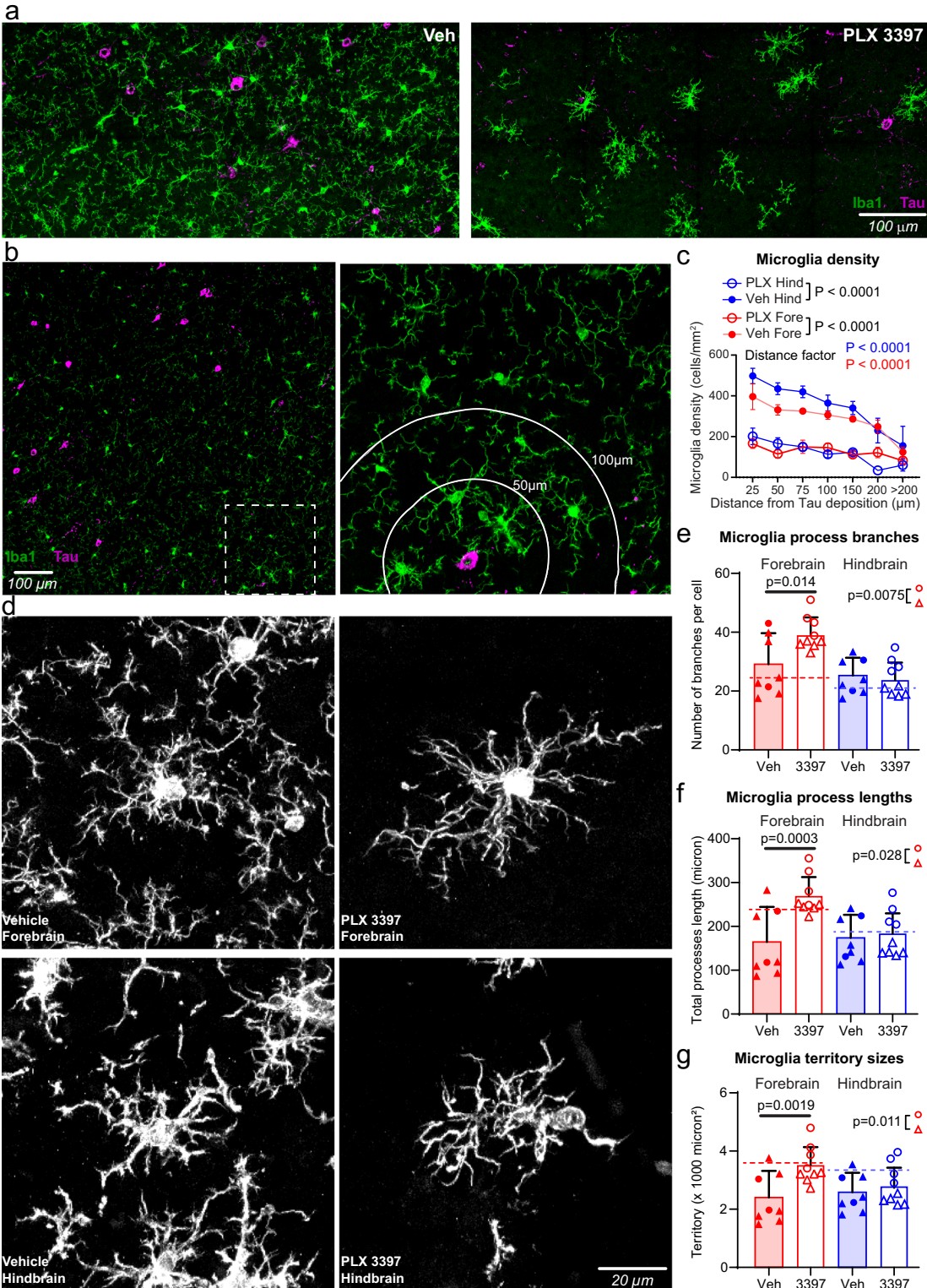

**Fig. 9 | PLX3397 treatment preferentially eliminates reactive microglia around tau deposits. a** Representative confocal images of immunostaining of tau protein (magenta) and microglia (Iba1, green) in the brain of a vehicle-treated (left) or PLX3397-treated (right) 210-d-old Tg2541 mouse. **b** Representative confocal images, similar to panel a, of a vehicle-treated 210-d-old Tg2541 mouse. The right panel shows the zoomed image from the dashed box in the left panel. The white lines in the right panel show distances from the tau deposit at the center. **c** Quantification of microglial densities at different distances from the nearest tau deposit in Tg2541 mice treated with vehicle or PLX3397 (275 mg/kg oral). Data are presented as mean ± S.E.M. Two-way ANOVA was used to compare statistical differences between treatment groups and distance bins. **d** Representative confocal images of microglial processes labeled by Iba1 immunohistochemistry. **e**–**g**, Quantification of microglial processes branch numbers (**e**), total lengths (**f**) and territory sizes (**g**) in Tg2541 mice treated with vehicle or PLX3397. Each data point shows the average value of the microglia measured in an individual mouse. Error bars represent the s.d. of the mean. Circles indicate female mice and triangles indicate male mice. Dotted lines show the average measurements from microglia in wild type mice. Mann–Whitney tests were used to compare between groups. The forebrain regions of PLX3397-treated male and female mice were also compared directly and the *P* values are shown on each plot. In (**c**), *n* = 10 mice for PLX groups and *n* = 8 mice for Veh groups. In (**e**–**g**), *n* = 9 mice for PLX groups and *n* = 8 mice for Veh groups. Source data are provided as a Source Data file.

unknown, and thus it remains unclear how kinase inhibition in OPCs by PLX compounds contributes, if at all, to the mechanism of action. Nevertheless, this topic warrants further investigation.

Tau pathology in Tg2541 mice is associated with moderate microgliosis and an upregulation of transcriptomic signatures of microglial activation[10,50]. Our transcriptome analysis showed activation of two major clusters of microglial-specific genes in Tg2541 mice. These two clusters had a high degree of overlap with the immune activation and suppression modules recently described in tauopathy mice and FTLD patients[10], indicating a specific reactive transcriptional program of microglia towards tau pathology. Consistent with this view, genes in the activated clusters also matched transcriptome modules described in activated microglia in neurodegeneration models (such as *Itgax* and *Clec7a*), but not in tumor or acute inflammation models[50]. We found an additional cluster of microglial genes that had similar expression in Tg2541 and wild type mice. Intriguingly, this cluster was not affected by PLX treatment in female mice, while in male mice this cluster exhibited marked activation (Fig. 10). Considering that PLX eliminates more than half of the total microglial population, a parsimonious explanation for this sustained gene cluster is that they are preferentially expressed by an inert microglial subpopulation that does not respond to tauopathy or low-dose CSF1R inhibition[61]. Consistent with this notion, we found that PLX treatment preferentially eliminates activated microglia in the vicinity of tau deposits, and thus most surviving microglia are not in direct contact with tau-laden neurons (Fig. 9). This is in contrast to Aβ mice, in which the surviving microglia are usually associated with Aβ plaques following PLX treatment[22,62]. We found that surviving microglia in female mice were non-inflammatory, and had longer and more elaborate processes compared to vehicle-treated microglia, showing functional and morphological features more similar to those of wild type microglia (Fig. 9). In sum, our data describe a microglial genetic signature that remains stable in Tg2541 mice with or without PLX treatment, likely representing a "dormant" microglial subpopulation that are less dependent on CSF1R for survival, or are less sensitive to CSF1R inhibition at the doses administered in our study.

Sex-specific differences exist in mouse microglial morphology, function, gene expression, and response to tauopathy, and the differences increase with age[63-66]. Our data identify a sex-dependent effect on therapeutic exposure and efficacy of CSF1R inhibition in Tg2541 mice. A difference in the plasma levels of PLX3397 during *ad libitum* oral dosing in male and female mice has been noted previously[17]; however, only male mice were evaluated further. We examined both male and female Tg2541 mice and found that, despite similar food intake, plasma and brain levels of PLX3397 were higher in male mice compared to female mice (Fig. 3 and Supplementary Fig. 9). However, at this level of drug exposure, only female mice received a functional benefit from CSF1R inhibition, an unexpected and clinically relevant outcome that would have been overlooked had our analysis been focused on a single sex. In male Tg2541 mice, despite a robust reduction of microglia and pathogenic tau, PLX treatment did not slow weight loss or extend survival, and plasma NfL levels were significantly increased, indicative of neuronal damage[41]. Neurodegeneration in Tg2541 mice has been shown to be largely limited to spinal cord motor neurons[23,67], and we found that NeuN and total tau immunoreactivity in the brain were largely unchanged by CSF1R inhibition (Supplementary Fig. 8). Thus, the functional deficits we measured are likely caused by neuronal dysfunction rather than neuronal death, but can be rescued by CSF1R inhibition. It has been suggested before that microglia from male animals may exhibit an increased responsiveness to CSF1R depletion compared to microglia from female animals[68]. Our results indicate that despite robust on-target effects for microglial depletion, male mice developed a PLX-induced inflammatory microglial phenotype (Fig. 10). Furthermore, we observed a concentration-dependent activation of IEGs in PLX treated Tg2541 mice, suggesting that

excessive PLX dosing in male mice may lead to excitotoxicity (Fig. 6), thus masking the beneficial effect of tau reduction. Microglia in male mice adopted a transcriptome and morphological phenotype that have been previously linked to excitotoxicity[54,69]. Curiously, IEGs were not significantly upregulated in male wild type mice, indicating that their activation may not be due to high concentration of PLX alone, but may also be dependent on tau deposits. Consistent with this premise, we identified increased glutamate and GABA neurotransmitter levels in the brains of male Tg2541 mice, but not male wild type mice or any female mice (Fig. 7), indicative of excitotoxicity following PLX dosing. Previous studies have linked tau accumulation and aberrant neural activity in vivo[70]. On the other hand, microglia are known to mediate neuroprotection against excitotoxicity[69,71] and elimination of microglia can exacerbate seizures and related neuronal degeneration[72]. Therefore, the concurrent tau removal and drug-induced inflammation driven by surviving microglia may increase the risk for hyperactivity, resulting in excitotoxicity in male mice with high PLX concentrations. Other sex-specific differences may also contribute to microglial sensitivity to CSF1R inhibition by a currently unknown mechanism. Future translational studies of pharmacological CSF1R inhibitors will need to carefully evaluate the role of sex in both safety and therapeutic outcomes.

CSF1R inhibitors were shown to be protective in mouse models of other neurodegenerative diseases, such as AD and Down syndrome[22,73,74]. However, under different treatment conditions, CSF1R inhibition did not affect Aβ plaque burden, but did rescue some functional deficits[62,75]. Therefore, microglia play a dynamic role in the brain's response to Aβ pathogenesis, and their attenuation may impart distinct benefits at different stages of disease. Our results suggest that, in primary human tauopathies, a subset of microglia play a net negative role before, during, and after disease onset and that their removal may be a viable therapeutic strategy. It remains to be determined if similar benefits should be expected for tauopathy in AD given the preceding comorbid Aβ pathology, but this may be elucidated in rodent models co-expressing human tau and Aβ. Nevertheless, because CSF1R inhibition has not been reported to be detrimental in Aβ mice, CSF1R inhibitors could ameliorate AD-related tauopathy even if caused by different disease mechanisms. Ongoing human clinical trials of CSF1R inhibitors in AD (e.g., NCT04121208) may provide additional mechanistic insights.

Primary human tauopathies constitute a class of neurodegenerative diseases caused by tau misfolding and aggregation and include progressive supranuclear palsy (PSP), corticobasal degeneration (CBD) and Pick's disease, among others. When combined with AD, in which tau aggregation follows Aβ deposition, tauopathies afflict a significant proportion of the human population, and thus novel approaches to directly or indirectly block tau pathogenesis or its downstream effects are urgently needed. Our study highlights several aspects of pharmacological CSF1R inhibition that bolster its therapeutic potential for human tauopathies. First, we observed greater efficacy of early (acute) CSF1R inhibition to restrict tau-prion levels in forebrain regions (Fig. 1e), likely due to reduced disease severity there relative to hindbrain regions. At later stages (chronic and terminal), CSF1R inhibition with PLX3397 did reduce tau-prion levels in the hindbrain, albeit a modest reduction relative to the effects in the forebrain (Fig. 1f, g). These findings suggest that early and long-term CSF1R inhibition (though not necessarily continuous) would most effectively mitigate human tauopathy. Second, we demonstrated that interventional dosing of Tg2541 mice, initiated at a stage when robust tau deposition had already occurred[23,27], led to a significant reduction in pathogenic tau (Supplementary Fig. 4). Therefore, our data support some potential clinical benefit of CSF1R inhibitors for treatment, in addition to prevention, of tauopathy. This is important because prophylactic prevention of non-autosomal dominant neurodegenerative diseases may be difficult due to a lack of definitive prognostic biomarkers paired

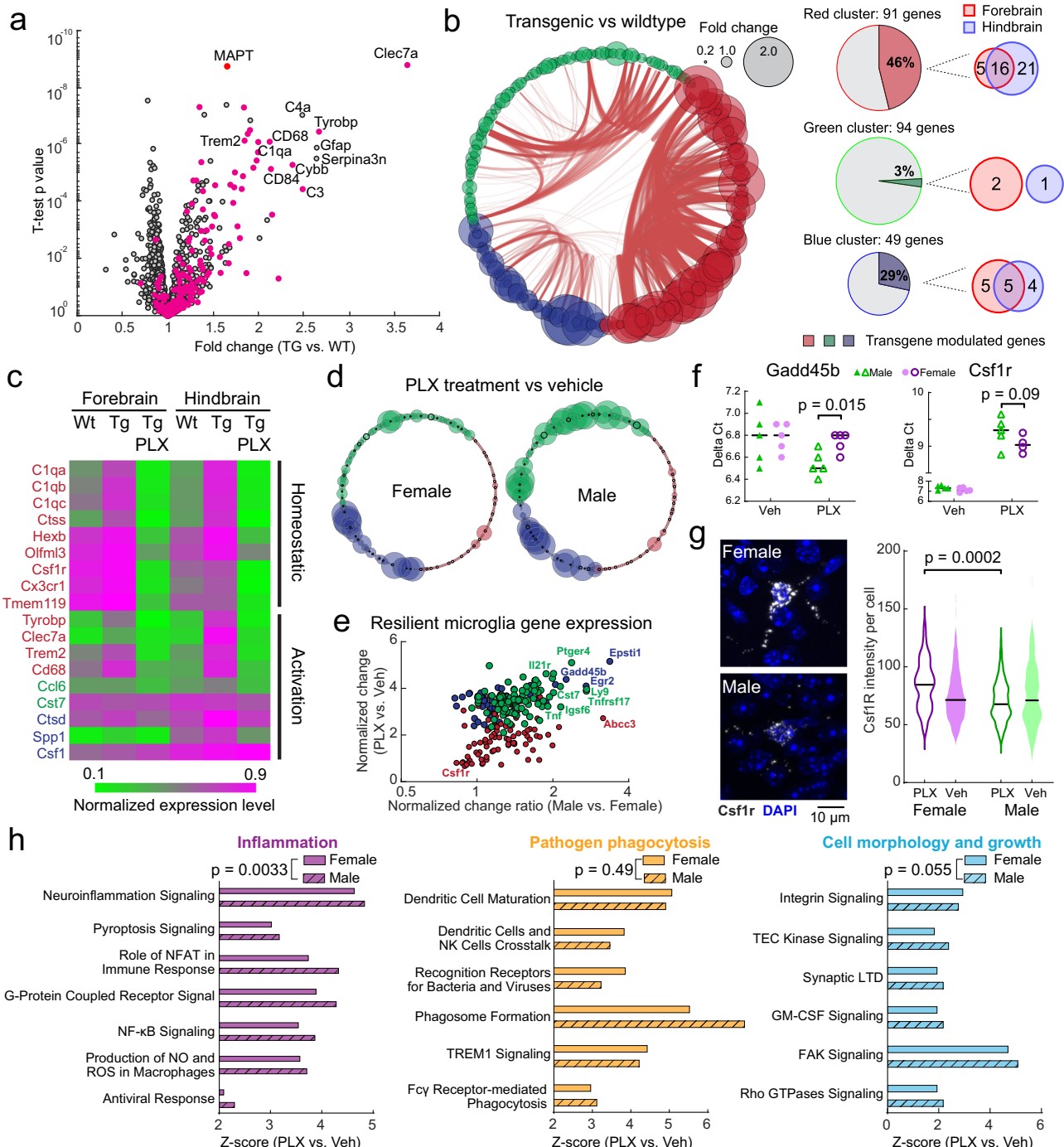

**Fig. 10 | Selective ablation of tau-activated microglial gene expression by PLX5622. a** Volcano plot of gene expression changes between Tg2541 and wild type mice. Many microglial-specific genes (magenta dots) show trends of up-regulation in Tg2541 mice. Two-sided *t*-tests were used to calculate *P* values. *P* values were not corrected for multiple comparisons. **b** Schemaball graphs of microglial-specific genes comparing Tg2541 to wild type. Individual circle sizes indicate fold changes. Microglial-specific genes (242 genes in our dataset) are clustered into three groups: red, green and blue. Connecting arcs between genes represent the degree of correlation. The pie charts on the right show the percentage of genes in each group that are differentially expressed in Tg2541 and wild type mice, and the Venn diagrams indicate the brain regions of differential expression. **c** Heatmap showing expression level changes in DAM genes[51]. Gene names are color coded to show their group assignments. **d** Schemaball graphs of microglial-specific genes comparing Tg2541 vehicle to PLX5622 treatment (1200 mg/kg oral) in male and female mice.

**e** Estimated gene expression levels in resilient microglia comparing between sexes and treatment. Each dot represents a different gene, and is color-coded according to the cluster to which it belongs. **f** Real-time PCR quantification of example microglial genes *Gadd45b* and *Csf1r* comparing vehicle to PLX5622 treatment in Tg2541 mice. Bars indicate medians and student's *t*-tests were used to compare groups. *N* = 5 mice for each group. One-sided *t*-tests were used for comparison. **g** Representative images and quantification of *Csf1r* gene expression by in situ mRNA hybridization (RNAscope) analyses of Tg2541 mice treated with PLX3397 from 2 mo of age until death. *n* = 5 mice per sex and treatment group. Data were quantified from 15 images collected in the cortex from each mouse. Bars indicate medians and two-way ANOVA was used to compare groups. Two-sided *t*-test was used for comparison. **h** Ingenuity Pathway Analysis of the gene expression patterns in the resilient microglia in male and female mice. Paired *t*-tests were used in each category to compare sexes. Source data are provided as a Source Data file.

with the fact that aggregation of the causative proteins can occur years or decades prior to symptom onset[30]. Third, we found that intermittent dosing of Tg2541 mice at three-week intervals produced a significant reduction in pathogenic tau (Supplementary Fig. 4). Despite relatively minimal off-target effects from continuous, long-term dosing of CSF1R inhibitors in mice[22], non-human primates[76], and humans[21], intermittent dosing would be clinically preferable if a similar therapeutic outcome was achieved, given the important functions for microglia and related peripheral cells in innate immunity. Because neurodegenerative tauopathies are slow, protracted diseases and microglia are long-lived[77], it is conceivable that breaks in dosing may occur on the order of months or years and be informed by medical imaging probes for microglia activation[78]. Imaging will be highly valuable for guiding intermittent dosing because microglial repopulation in a pathological setting (e.g., disease or normal aging) may not necessarily result in a return to baseline transcriptional profiles. Fourth, we found CSF1R inhibition to extend the survival of female Tg2541 mice (Fig. 2), indicating that the reduction in pathogenic tau in this model system translates to an improved clinically relevant outcome. We postulate that if CSF1R inhibitor dosing was optimized for male Tg2541 mice, any adverse effects in the CNS or periphery would likely be diminished and their survival extended. Fifth, we showed that complete microglial depletion is not necessary, or even desirable, for a therapeutic benefit. As discussed above, the microglia that survive CSF1R inhibition represent a unique microglial sub-population that likely serves important functions in brain homeostasis. Future preclinical studies may pinpoint the precise level, timing, and frequency of CSF1R inhibition such that the detrimental effects of microglial activation are minimized while an appropriate number of homeostatic microglia remain for brain surveillance. Lastly, CSF1R inhibitors applied in conjunction with tau immunotherapy may prove to be a successful combination therapy; because microglia are not needed for antibody effector function[79], removing tauopathy-activated microglia would slow tau pathogenesis and may also increase the efficacy of tau immunotherapy. Taken together, our data support the therapeutic modulation of microglial activation by CSF1R inhibitors as a potential approach to treating human tauopathies.

## Methods

### Animals

Animals were maintained in a facility accredited by the Association for Assessment and Accreditation of Laboratory Animal Care International in accordance with the Guide for the Care and Use of Laboratory Animals. All procedures for animal use were approved by the University of California, San Francisco's Institutional Animal Care and Use Committee under protocol #AN182216-03. Mice were housed in standard ventilated cages with food and water provided ad libitum. The Tg2541 transgenic mouse line expresses the human 0N4R tau isoform under the Thy1.2 genetic promoter. Tg2541 mice (a kind gift from Dr. Michel Goedert) were originally generated on a mixed C57BL/6 J × CBA/Ca background[23] and were then bred onto a congenic C57BL/6 J background using marker-assisted backcrossing for eight generations before intercrossing to generate homozygous mice. Albino Tg2541 mice were generated by intercrossing Tg2541 with C57BL/6 J mice expressing a spontaneous mutation in the tyrosinase gene (homozygous for Tyr$^{c-2J}$) causing albinism (Jackson Laboratory; 000058). To generate mice for in vivo bioluminescence imaging, we employed Tg(*Gfap*-luc) mice, which express firefly luciferase under the control of the murine *Gfap* promoter (a kind gift from Caliper Life Sciences). These reporter mice were originally on the FVB background, but were backcrossed to a congenic B6 background, and then crossed to the B6-albino background. To create bigenic mice, albino Tg2541 mice were crossed with albino Tg(*Gfap*-luc) animals to produce double hemizygotes. Next, double hemizygotes were crossed and the progeny were screened for the presence of both transgenes expressed at homozygosity.

### PLX compound formulation in mouse chow

PLX3397 was provided by Plexxikon Inc. and was formulated in AIN-76A standard chow by Research Diets Inc. at 275 mg/kg[80]. PLX5622 was provided by Plexxikon Inc. and was formulated in AIN-76A standard chow by Research Diets Inc. at 1200 mg/kg[62]. PLX73086 was provided by Plexxikon Inc. and was formulated in AIN-76A standard chow by Research Diets Inc. at 200 mg/kg as recommended by Plexxikon Inc.

### Immunohistochemistry and slide scanning

Formalin-fixed samples were embedded into paraffin (FFPE) using standard procedures and microtome-cut into 8 μm sagittal brain sections or coronal spinal cord sections and mounted onto slides. To reduce tissue autofluorescence, paraffin slides were photobleached for 48 h. Slides were deparaffinized in a 61 °C oven for 15 min and rehydrated through alcohols. Antigen retrieval was performed by autoclaving for 10 min at 121 °C in 0.01 M citrate buffer. Sections were blocked in 10% normal goat serum (NGS) (Vector Labs) for 1 h at room temperature. Primary antibodies included rabbit monoclonal anti-Iba1 (Abcam, ab178847, 1:250), rabbit polyclonal anti-P2yr12 (Atlas, HPA014518, 1:250), mouse monoclonal anti-NeuN (Millipore, MAB377, 1:250), chicken polyclonal anti-GFAP (Abcam, ab4674, 1:250), mouse monoclonal anti-pS202/T205 tau (AT8; Thermo Fisher, MN1020, 1:250), and rat monoclonal CD206 (Biorad, MCA2235, 1:250). Primary antibodies were diluted in 10% NGS in PBS and allowed to incubate on the slides overnight at room temperature. Primary antibody detection was performed using goat anti-mouse, rabbit, or chicken IgG (H + L) highly cross-adsorbed secondary antibodies conjugated to Alexa Fluor Plus 488, 555, or 647 (Life Technologies) at 1:500 dilution. Slides were cover-slipped using PermaFluor mounting medium (Thermo). Whole-section tiled images were acquired with an Axioscan.Z1 slide scanner (Zeiss) at 20× magnification, and quantification was performed with Zen 2.3 software (Zeiss).

### Cellular bioassay to measure tau-prion activity

A HEK293T cell line (ATCC, CRL-3216) expressing the repeat domain of 4 R human tau (aa 243–375) containing the P301L and V337M mutations and C-terminally fused to YFP was generated previously[29]. A stable monoclonal line was maintained in DMEM, supplemented with 10% FBS and 1% penicillin/streptomycin. To perform the bioassay, 3,000 cells (containing 0.1 μg/ml Hoechst 33342) were plated in 70 μl per well into 384-well plates (Greiner) and incubated for 2 h before treatment with samples. Clarified brain lysate at a final concentration of 1.25 μg/mL total protein was first incubated with Lipofectamine 2000 (0.2% final concentration) and OptiMEM (9.8% final concentration) for 90 min, and then added to the plated cells in quadruplicate. Plates were incubated at 37 °C for 1–3 days, and then the live cells were imaged using an INCell Analyzer 6000 Cell Imaging System (GE Healthcare) and custom algorithms were used to detect fluorescent YFP-positive puncta (aggregates).

### Mechanical tissue homogenization

Postmortem brains and spinal cords were thawed and weighed to determine the mass in grams. Brains were bisected into forebrain and hindbrain pieces using a single cut with a scalpel blade between the striatum and hypothalamus. Tissue was homogenized in nine volumes of cold DPBS containing Halt Protease Inhibitor Cocktail (1x, Thermo Fisher Scientific) using a Precellys 24-bead beater (Bertin Instruments) with metal bead lysing matrix (MP Biomedical). Where necessary, brain lysates were clarified by centrifugation at 10,000 × g for 10 min at 4 °C. All tissue and samples were stored at −80 °C until further use.

## Formic acid extraction of insoluble proteins in brain tissue for ELISA

Fifty microliters of formic acid were added to 25 μL of 10% brain homogenate and placed in an ultracentrifuge tube. The samples were vortexed, sonicated for 20 min at 37 °C in a water-bath sonicator, and then centrifuged at $100,000 \times g$ for 1 h. Fifty microliters of supernatant were recovered to a low-binding tube and neutralized with 950 μL of neutralization buffer (1 M Tris base and 500 mM dibasic sodium phosphate). Samples were aliquoted into low-binding tubes and flash frozen in liquid nitrogen. The following ELISA kits from Thermo Fisher Scientific were used according to the manufacturer's protocols: total tau (KHB0041), p-tau S396 (KHB7031), and p-tau T231 (KHB8051). Each sample was analyzed in duplicate. Raw ELISA values were adjusted to total brain protein (grams) in the clarified 10% brain homogenate as determined by bicinchoninic acid (BCA) assay (Pierce/Thermo Fisher Scientific).

## Quantification of total protein in brain homogenate

Total protein content in the PBS-soluble (clarified 10% brain homogenate) and detergent-soluble fractions was quantified using the Pierce BCA Protein Assay Kit (Thermo Fisher Scientific) following the manufacturer's protocol.

## Generation of tau K18*P301L fibrils

Production, purification and fibrillization of recombinant tau K18*P301L fibrils were performed as previously described[81]. Briefly, lyophilized peptides were dissolved in DPBS at 145 μM and incubated for 72 h at 37 °C under constant agitation at 900 rpm.

## Stereotaxic injections in Tg2541 mice

Forebrain inoculation: Ten-week-old Tg2541 mice received unilateral inoculations of 10 μl of 1.5 mg/ml tau K18 P301L fibrils using stereotaxic methods. Injections followed a two-step process: the needle was first advanced to the hippocampus (Bregma −2.5 mm, Lateral 2.0 mm; Depth −2.3 mm from the skull surface) to deliver 5 μl over three minutes, then the Hamilton syringe pump was paused for five minutes to allow for diffusion prior to retracting the needle to the overlying cortex (Depth −1.3 mm) where the remaining 5 μl was injected. After fibril injection, the needle remained in place for five minutes to allow for diffusion of fibrils before retraction, patching the skull and suturing the scalp. Midbrain inoculation: Ten-week old Tg2541 mice received bi-lateral inoculations of 10 μl of 1.5 mg/ml tau K18 P301L fibrils using stereotaxic methods. Five microliters was injected at each site in the midbrain (Bregma, −4.3 mm; Lateral, 1.0 mm, Depth, −2.5 mm) and (Bregma, −4.3 mm; Lateral, −1.0 mm, Depth, −2.5 mm).

## Automated home cage monitoring of behavior

Total activity measurements of freely moving mice were made every 30 days after PLX dosing in Promethion cages (Sable Systems International). At each time point, mice were first randomized and placed individually in Promethion cages for 4 to 6 days. Real-time cage activity recording was continuous during the entire session using a combination of a running wheel with sensors to measure speed and distance traveled, three balances to measure body weight, food and water consumption, and a matrix of infrared light beams to measure XYZ movements with 0.25 cm resolution. Analysis of these metrics was used to detect behaviors such as sleep, rearing and general locomotion. For each mouse, data used for analyses were average readings per light or dark cycle. Data from the first circadian cycle were excluded due to variable behavior during habituation. To calculate the activity scores, wheel use, locomotion and rearing were first normalized to a 0–1 scale by the maximum value in the whole dataset, and then the geometric mean of the normalized values for each session was calculated.

## Quantification of PLX compound levels in brain tissue and plasma

Brain homogenates (20% w/v) were prepared in PBS by one 30-second cycle of bead beating at 5500 rpm with a Precellys 24-bead beater (Bertin Instruments) or plasma samples were prepared by dilution to 25% with PBS. Compounds were recovered by mixing equal parts of brain homogenate with a 50/50 (v/v) solution of acetonitrile (ACN) and methanol containing 1 mM niflumic acid. Precipitated proteins were removed by vacuum filtration (Captiva ND, Agilent). Analysis was performed using a liquid chromatography-tandem mass spectrometry system consisting of an API4500 triple quadra-pole instrument (AB Sciex, Foster City, CA) interfaced with a CBM-20A controller, LC20AD 230 pumps, and a SIL-5000 auto-sampler (Shimadzu Scientific, Columbia, MD). Samples were injected onto a BDS Hypersil C8 column maintained at room temperature. The amount of ACN in the gradient was increased from 75–95% ACN over two minutes, held for one minute, and then re-equilibrated to 75% ACN over 1.4 min. Data acquisition used multiple reaction monitoring in the positive ion mode. Specific methods were developed for each compound (PLX3397 and PLX5622), enabling the determination of absolute concentrations.

## Blood plasma neurofilament light (NfL) protein measurement using SIMOA

At monthly time points, 150 μl blood was collected in EDTA-coated tubes. The plasma was centrifuged at $1000 \times g$ for ten minutes to clarify the samples, and was then diluted with sample diluent buffer included in the kit by 25-fold and 100-fold, respectively, prior to the measurement. Plasma NfL concentration was measured and analyzed using the NfL kit (Quanterix) with the SIMOA HD-1 analyzer (Quanterix). Briefly, samples, magnetic beads coated with capture antibody, and biotinylated detector antibodies were combined. Thereafter, the capture beads were resuspended with streptavidin-β-galactosidase (SBG) and resorufin β-D-galactopyranoside (RGP) and transferred to the SIMOA disk. Each bead fit into a microwell in the disk and if NfL was captured then the SBG hydrolysed the RGP substrate which generated a fluorescent signal, and then the concentration was measured against a standard curve derived from known concentrations of recombinant NfL included in the kit. The lower limit of quantification of the assay for plasma was 17.15 pg/mL.

## Masson's trichrome and Picosirius red staining

FFPE liver sections (8 μm) were deparaffinized through xylenes and graded alcohols and then rehydrated in distilled water. The Masson's trichrome staining kit (Abcam #ab150686) was used according to the manufacturer's protocol. The Picro Sirius Red Stain Kit (Abcam #ab150681) was used to stain tissue sections for 60 min at room temperature. The slides were rinsed in two changes of acetic acid, three changes of ethanol, and then mounted using PermaFluor mounting medium (Thermo). Whole-section tiled images were acquired with an Axioscan.Z1 slide scanner (Zeiss) at 20× magnification, and quantification was performed with Zen 2.3 software (Zeiss).

## Alkaline phosphatase (ALP) ELISA of plasma samples

Mouse plasma samples were diluted 1:100 in the provided dilution buffer and measured using the ALP ELISA kit (Biovision #E4572-100) according to the manufacturer's protocol.

## RNA extraction and Nanostring RNA expression measurements

RNAlater-preserved samples were homogenized in PBS and total RNA was extracted from samples using the Quick-RNA Miniprep Kit (Zymo Research). RNA extracts were evaluated for concentration and purity using a Nanodrop 8000 instrument (Thermo Fisher Scientific) and diluted to a concentration of 20 ng/μl. Hybridizations were performed for the mouse Neuroinflammation, Myeloid cell, and Neuropathology panels according to the nCounter XT Assay user manual (Nanostring).

The hybridizations were incubated at 65 °C for 16 h, and then were added to the nCounter SPRINT Cartridge for data collection using the nCounter SPRINT Profiler. Counts were analyzed using the nSolver Analysis Software.

## RNA expression analysis

In total, there were 10 mice in the Tg2541 vehicle group, 10 mice in the Tg2541 PLX5622 group, 6 mice in the wild type vehicle group, and 6 mice in the wild type PLX5622 group. Each mouse had separate forebrain and hindbrain samples and three panels of Nanostring sequencing were performed on each sample. Data from the three panels were pooled together to form the final dataset. When pooling data, if a gene appeared in more than one panel then the average read value was used in subsequent analysis, unless one panel failed to detect the gene.

To assign cell-type specificity of each gene, we used the transcriptome dataset reported in a previous study[82], inspired by previously reported approaches in bulk tissue samples[50]. We set a specificity threshold in which a gene qualified to be cell-type specific if its expression in a cell type was greater than five times the sum in all other cell types. Using this standard, our dataset had 242 microglial-specific genes, 47 astrocyte-specific genes and 70 neuron-specific genes. All cell type specific gene analyses were repeated with a three-time threshold.

We used partial least-square (PLS) regression (MATLAB) to extract the gene expression pattern aligned with Tg2541-wild type axis, using individual gene reads from each mouse as predictors and genotype as responses. Only vehicle groups were used in constructing the PLS regression. Forebrain and hindbrain were calculated separately. Five output dimensions were chosen for all PLS analyses, as they covered 99.99% of the total variance in all cases. The scores in the first two dimensions were plotted. To project PLX3397-treated groups to the PLS dimensions, we used the following formula:

$$Score_{projection} = (Loading_{predictor} \backslash (raw - mean_{predictor})')' \quad (1)$$

To calculate population vector distance, we use the "mhal" command in MATLAB. All five dimensions were used for each mouse. The wild type vehicle group was used as a target.

To calculate the vector angle, each mouse's gene expression pattern was regarded as a five-dimension vector in the PLS space, and the angle between each mouse and the average vector of the wild type vehicle group was calculated with the following formula:

$$A = \cos^{-1}((u \cdot v)/(|u||v|)) \quad (2)$$

To calculate the PLS regression along the PLX concentration and sex-correlated dimensions, we constructed regressions using all non-microglial genes or only immediate early genes[83] to measured brain PLX concentrations and sex of each sample. We then calculated variable importance in projection to isolate the genes important for the regression. To calculate the projected PLX concentrations, we used the products of gene expression levels and coefficients estimated from PLS regression.

To calculate clusters in the microglial-specific genes, we calculated pairwise Pearson's correlation coefficients across 32 samples among each gene. The resulting similarity matrix was then processed with a generalized Louvain community detection algorithm[52].

To estimate the gene expression levels of the resilient microglia, we used 6 genes (Tmem119, P2ry12, Fcrls, Olfml3, Itgam v1, and Itgam v2) as microglial-specific house-keeping genes[84] and used their levels relative to the vehicle-treated group to scale the expression levels of other microglial genes. We used these estimated expression levels as input for the Ingenuity Pathway Analysis (QIAGEN).

## Gene expression analysis by RT-qPCR

Mouse brains were collected at endpoints and flash frozen in DNA/RNA shield reagent. Tissue was homogenized as described above and total RNA was purified using a commercial isolation kit (Zymo Research). RNA concentration and the RNA integrity number (RIN) were determined using a Bioanalyzer 2100 instrument and an Agilent RNA 6000 Pico Kit (Agilent 5067-1513). Only samples with a RIN score ≥7.0 were used for gene expression analysis. To confirm transcriptome profiling results, 2.5 ng of sample mRNA was applied to triplicate RT-qPCR reactions consisting of 1x TaqPath 1-Step Multiplex Master Mix (ThermoFisher Scientific A28526), Taqman primer/probe sets and a normalizing human MAPT Taqman assay. Reactions were run on a QuantStudio 6 and 7 Pro instrument and amplification yielding cycle threshold ($C_T$) values were corrected with Mustang Purple passive reference dye for each target gene. Gene expression of PLX-treated mice relative to vehicle-treated mice was determined using the following comparative $C_T$ equation and values were expressed as fold-change:

(3) $2 - \Delta\Delta C_T = [(C_T$ gene of interest $- C_T$ hMAPT internal control)] PLX-treated mice $- [(C_T$ gene of interest $- C_T$ hMAPT internal control)] vehicle-treated mice.

## In situ mRNA hybridization (RNAscope) analysis

All RNAscope experiments were performed using FFPE tissue sections of age-matched male and female mice from transgenic tau and wild type control lines. The RNAscope Multiplex Fluorescent V2 Reagent Kit (#323100) was used according to the manufacturer's instructions, other than the target retrieval incubation step, which was carried out at 95 °C for 20 min. The following probes, all from ACD Bio (Newark, CA), were used: *MAPT* (#522491), *Fos* (#506931-C2), and *Csf1r* (#428191-C3).

## HILIC-MS/MS metabolomics analysis

HILIC-MS/MS analysis was carried out by the West Coast Metabolomics Center at the University of California, Davis. Briefly, four milligrams of mouse cortex were homogenized with 3.2 mm stainless steel beads using a bead-beater in 750 µl of methyl-tertiary butyl ether and 225 µl of methanol. Samples were then vortexed for 10 s, shaken on an orbital shaker at 4 °C for 5 min, and then 188 µl of liquid chromatography mass spectrometry (LCMS)-grade water was added. Samples were vortexed again for 20 s and centrifuged for 2 min at 14,000 × g. 110 µl of the lower phase (all polar metabolites) were then moved to a new 1.5 ml tube and dried using a centrivap. Dried samples were resuspended in 100 µL buffer containing internal standards and then analyzed using an Agilent 1290 ultra-high pressure liquid chromatography (UHPLC)/Sciex Triple-time-of-flight (TOF) 6600 mass spectrometer. Briefly, a Waters Acquity Premier UHPLC bridged ethylene hybrid (BEH) Amide Column (1.7 µm, 2.1 × 50 mm) was used with a mobile phase A of ultrapure water with 10 mM ammonium formate (AF) and 0.125% formic acid (FA) (pH 3.0), and a mobile phase B of 95:5 (v/v) acetonitrile:ultrapure water with 10 mM AF + 0.125% FA (pH 3.0). A column temperature of 45 °C and a gradient of 0 min, 100% B; 0.5 min, 100% B; 1.95 min, 70% B; 2.55 min, 30% B; 3.15 min, 100% B; 3.8 min, 100% B, were used with a flow rate of 0.8 ml/min.

Following data collection, internal standard chromatograms were examined for consistency of peak height and retention time as quality control measures. Raw data files were then processed using an updated version of the MS-DIAL software[85] using an in-house mzRT library and MS/MS spectral matching with NIST/MoNA libraries. All MS/MS annotations were then manually curated to ensure that only high-quality metabolite identifications were included.

## In vivo bioluminescence imaging

Bioluminescence imaging was performed on the brains and spinal cords of albino bigenic Tg(2541:*Gfap*-luc) homozygous mice after receiving an intraperitoneal injection of 25 mg/kg cyclic luciferin-1 (CycLuc1) sodium salt solution (Aobious; AOB6377) prepared in PBS,

**Article** https://doi.org/10.1038/s41467-022-35753-w

pH 7.4. After CycLuc1 injection, mice were placed in an anesthetization chamber and exposed to an isoflurane/oxygen gas mix for ten minutes. During this time, the heads of the mice were shaved to enhance the bioluminescence signal. After anesthetization, mice were placed in an IVIS Lumina III small animal imaging system (PerkinElmer) and were kept under constant anesthesia. Mice were imaged for 60 s duration at three time points (14, 16 and 18 min) following CycLuc1 injection as determined in one-hour time-lapse calibration studies. After image acquisition, the mice were allowed to recover in their home cages. Brain and spinal cord bioluminescence values were calculated from images displaying surface radiance using standardized regions of interest and were then converted to total photon flux (photons per second) using Living Image software version 4.4 (PerkinElmer).

## Confocal imaging of thick tissue sections

Vibratome-sectioned brain slices (40 μm thick) were immunolabeled with Iba1 (1:250) and AT8 (1:250) antibodies, using standard protocols for free-floating sections in multi-well plates. Sections were mounted using PermaFluor and #1.5 coverglass. Using a Leica SP8 confocal microscope equipped with HyD detectors and an AOBS, samples were first visualized using Navigator function to acquire an overview image of each slice using a 20× water-immersion lens (0.95 NA). From the mosaic image, smaller tiled-ROIs were marked in the forebrain and hindbrain to acquire high-resolution, sequential-scanned image stacks using a 63× water-immersion lens (1.2 NA). Eight-bit image z-stacks (1 μm steps) were collected at 512 × 512-pixel resolution. Images were processed using custom MATLAB code.

## Microglial morphology analysis

Microglial morphology was analyzed using a custom script in MATLAB. Briefly, raw confocal image stacks were smoothed and then maximally projected. Isolated microglia cells were manually selected for analysis. The selected microglia region was binarized with an intensity threshold, and then the cell body was detected by fitting a largest circle in the binary mask. After excluding the cell body region, the remaining microglia processes were skeletonized and branch number, branch length and bounding box were measured using "regionprops" and "bwmorph" commands.

## Statistical analysis

Statistical analyses were performed using GraphPad Prism 8. Comparisons between two groups were performed by unpaired $t$ test or by Mann-Whitney nonparametric test. For comparisons of more than two groups, one-, two-, or three-way ANOVA was performed with Holm-Šidák post hoc analysis. Following ANOVA, residuals were evaluated for normal distribution using the Anderson-Darling test and the data were evaluated for equal variance using the Brown-Forsythe test. If both assumptions were violated ($P < 0.05$), the data was reanalyzed using Welch's ANOVA with Dunnett T3 post hoc analysis. For repeated-measures ANOVA, sphericity was not assumed and the Geisser-Greenhouse correction was applied. If any data points were missing, a mixed-effects model (Restricted maximum likelihood; REML) was used instead. Pearson's correlation tests were performed as one-sided tests as, in each case, we had a directional hypothesis of either positive or negative correlation. All other statistical tests were performed as two-sided tests. Sample sizes are shown in graphs with each data point representing an individual mouse, or are reported in the figure legends. Experimental replication and exact statistical tests used are detailed in the figure legends. Throughout the manuscript, a.u. stands for arbitrary units.

## Reporting summary

Further information on research design is available in the Nature Portfolio Reporting Summary linked to this article.

## Data availability

The authors declare that all data supporting the findings of this study are available within the paper and its supplementary files. The transcriptome (Nanostring) data generated in this study, specifically in Figs. 6–9, have been deposited in the Zenodo database under accession code 7415371. Materials, data, code and associated protocols are promptly available upon request from the corresponding author. Source data are provided with this paper.

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

## Acknowledgements

We thank Plexxikon, Inc. for providing the PLX3977, PLX5622 and PLX73086 compounds, and Andrey Reymar and Brian West for consulting on drug dosing and chow formulation. We thank Julian Castaneda, Karina Walker and Lyn Batia (and their staff) at the UCSF Hunter's Point Animal Facility for coordinating transgenic mouse development, breeding, and drug efficacy studies. We thank Scott Mackell and the entire West Coast Metabolomics Facility (University of California, Davis) for performing the HILIC-MS/MS metabolomics analyses. We thank Louis Johnson (JMC Data Experts) for insightful discussions on the principal component analyses in our study. We thank Masahiro Inoue (Daiichi Sankyo, Inc.) for insightful discussions on pharmacokinetics and pharmacodynamics in our study. We thank Stanley Prusiner and David Ramsay at the UCSF Institute for Neurodegenerative Diseases (IND) for access to equipment and technical resources critical for the completion of this study. We thank the following UCSF IND staff for technical assistance: Abby Oehler, Rigoberto Roman-Albarran, Julia Becker, Marta Gavidia and Manuel Elepano. The study was funded by grants from the National Institutes of Health (# RF1 AG061874), the Rainwater Charitable Foundation, the Sherman Fairchild Foundation, the Henry M. Jackson Foundation, the Edward N. & Della L. Thome Memorial Foundation, and Daiichi Sankyo, Inc. P.Y. was funded by Shanghai Municipal Science and Technology Major Project and Shanghai Natural Science Foundation (22ZR1415000).

## Author contributions

C.C. conceived the study. N.R.J. and C.C. designed the experiments. N.R.J., E.C., T.P.L., W.Y., A.B., B.M.R., M.C.S., M.H., K.G., A.A., and C.C. performed experiments and prepared data. N.R.J., P.Y., and C.C. analyzed and interpreted data. N.R.J., P.Y., and C.C. wrote the paper. C.C. supervised the study.

## Competing interests

The Institute for Neurodegenerative Diseases (UCSF) had a research collaboration with Daiichi Sankyo, Inc. (Tokyo, Japan). All other authors declare no competing interests.
