## [Peer Review File · Nature Communications]

CSF1R inhibitors induce a sex-specific resilient microglial phenotype and functional rescue in a tauopathy mouse modelReviewers' comments:

Reviewer #1 (Remarks to the Author):

In this manuscript, Johnson et al investigate the impact of CSF1R inhibition-mediated microglia depletion in a tauopathy model. This is of importance as tauopathy is considered to contribute to neurodegeneration, for instance in the common neurodegenerative disorder Alzheimer's disease, yet we do not have approved therapies to intervene. CSF1R inhibition is a commonly used tool in experimental models to deplete macrophages, and inhibitors are being trialled in humans for treatment of glioma and Alzheimer's disease. However we still do not understand the differential impact of different CSF1R inhibitors, treatment regimes, or sex on outcomes. Here, the authors administer CSF1R inhibitors to a mouse model of tauopathy and make 3 major findings. First, there are differential impacts of CSF1R inhibition on reducing tau pathology depending on the brain region/severity of pathology, and treatment paradigm. Second, there are sex differences in impacts on extending survival (females) or proxies of neurodegeneration (males). Third, CSF1R inhibition doesn't eliminate all microglia, indicating there may be resistant vs susceptible populations in the context of tauopathy. The novelty is in comparison of treatment paradigms and sexes, building on previous work that administered CSF1R inhibitors to models of tauopathy only the male sex or investigated a single time point post treatment.

With regards to the data, I have the following suggestions.

Major suggestions:

1. I appreciate that the authors excluded impacts of the CSF1R inhibitors on peripheral macrophages (including monocytes which could then travel to the brain). However there are other CNS-resident macrophages (border associated macrophages), and these are known to be depleted by CSF1R. Of particular interest are the perivascular macrophages, which are directly adjacent to brain parenchyma. Can the authors assess the impact of their depletion paradigms on perivascular macrophage numbers (e.g. Lyve1+ IBA1+ cells, in association with blood vessels).
2. There are interesting suggestions throughout the paper suggesting impact on or rescue of neuronal damage – with NFL measured in the CSF, and the possibility of excitotoxicity is mentioned. As tauopathy is considered to be associated with neurodegeneration, can the authors assess neurodegeneration directly in the brain. Are there differences in the survival of neurons in the conditions where differences were observed, e.g. females with extended lifespans in PLX treated vs vehicle, males showing 'toxic' responses to high PLX blood/brain concentrations?
3. The two CSF1R inhibitors 5622 and 3397 are used interchangeably throughout the paper (i.e. both drugs are not used in every treatment paradigm), can the authors clarify why this is? It would be useful to at least have the quantification of microglia depletion for both drugs (as only 5622 is provided in the SFig1 but not for 3977) for IBA1 and P2RY12 measures, to be confident of experiments throughout the manuscript using 3977.
4. Given the focus on sex differences in the manuscript, it would be useful to see statistical comparisons between males and females for the 3 measures of tauopathy (as currently they are combined in Fig1).
5. Throughout the manuscript, it is not clear what the 'n' numbers in the graphs represent. Is every dot an individual mouse? Can this be clarified in the figure legend?
6. Can the authors compare the gene expression differences from the nanostring between males and females, females with longer lifespan from PLX vs vehicle, and males with increased toxicity or not? This could give indications as to why there are functional differences with the different groups following microglia depletion. Is there statistical significance between the male vs female groups for the IEGs?
7. Given the focus on the IEGs as a potential mechanism mediating differential responses to CSF1R, it would be useful to see validation of this at the qPCR or protein level. Ideally this would be on-tissue validation as induction of IEGs can occur in cells while tissue is being processed for sequencing.
8. Can the changes in astrocyte reactivity indicated by bioluminescence be validated by staining for GFAP in the tissue?
9. The authors suggest that the PLX drug toxicity in the male mice is due to off-target effects independent of microglia e.g. anemia, hepatotoxicity, but it wasn't clear to me that this is necessarily the case. Are there differences in neurodegeneration/neuronal health that are the cause? Do the mice actually show hepatotoxicity or anemia? It's quite important to know what

underpins the deleterious responses in males for translational relevance. Comparing the males that did poorly to the ones on the intermittent diet (where they did better) could be interesting.

Minor suggestions:

1. Can the authors discuss why there are some different responses seen between 3397 and 5622 with regards to impact on the 3 measures of tau pathology? Can the authors also discuss the implications of not all 3 measures being impacted in the same way even with the same drug and treatment regimen?
2. In supplemental figure 8, it looks like in the K18 fibril model the PLX treated males do indeed have an extended lifespan, which counters the sex specific conclusions from the paper. Can the authors clarify?
3. As the authors mentioned that microglia inflammasome activation can regulate tauopathy, can the authors check if inflammasome related genes were changed in the nanostring analysis with PLX treatment?
4. Can the authors change the title of figure 7 to more accurately explain what was done, as the current title referring to 'tau resistant' microglia is confusing and specific microglia subpopulations were not sorted then sequenced. Referring to wildtype vs transgenic bulk gene expression more accurately reflects what was done.
5. Can the authors include representative images of microglia depletion with PLX with respect to proximity to tau deposits (currently only the vehicle control images are shown).
6. Can the authors discuss – in the discussion – the implications for differences seen between forebrain and hindbrain responses in their model? What are the implications of human disease (regional responses, responses based on severity of tauopathy, etc).
7. The authors refer to DAMs as being known to have a pro inflammatory profile and be detrimental, this is not correct and needs to be amended. The DAM profile has been linked with anti inflammatory microglia profiles and even developmental and regenerative processes.

Reviewer #2 (Remarks to the Author):

The manuscript by Johnson et al. describes an interesting concept, connecting microglia and tauopathy and gender. Yet, the manuscript in its present form reads like a collection of partial stories that were not fully explored or integrated, and the conclusions are appealing but not fully supported and mechanistically clear.

Specific comments:

Figure 1:

The authors found that depletion of microglia had a different outcome on pathology, in different parts of the brain. Moreover, assessment by ELISA and by immunohistochemistry gave a different picture. Genders of the mice should be presented separately. In addition, it is puzzling why the early and brief depletion is more effective than continuous depletion, as evaluated by IHC. This experiment is missing visualization of the extent of depletion, and to what extent homing of myeloid cells from the periphery took place.

Figure 2:

The authors describe gender differences without any mechanistic explanation. Moreover, the presentation of the entire figure is confusing.

Figure 3:

Not all correlations with NFL shown in the figure add to the story.

Figure 5:

Gliosis should be measured by IHC.

Figure 6:

IB1 is not a valid marker for microglia, certainly not in a case in which microglia are partially depleted.

Reviewer #3 (Remarks to the Author):

The current study use a comprehensive treatment paradigms of CSF1R inhibitors to evaluate the roles of CSF1R inhibition in Tau-related pathology. They observed consistent reduction in the levels of pathogenic tau in the brains of both male and female Tg2541 mice with PLX treatment. Interestingly, only female mice benefited from extended survival, functional rescue, and reduced NfL levels. Hence, the authors raise an important issue to carefully evaluate the role of sex in the translational studies that target CSF1R inhibition. Overall the paper is descriptive and provides incremental knowledge to the field.

Comments:

1. The sex-specific differences could be attributed to different plasma and brain levels of PLX3397 between male and female mice. However, it remains unclear why the different drug levels did not impact its effects on reduction of microglia number and tau pathology.
2. The study did not have data demonstrating why the male mice tend to accumulate higher amounts of CSF1R inhibitors in their plasma and brain. Is it due to slower metabolism of the drug?
3. The study showed higher expression of immediate early genes (IEGs) in male mice which may be indicative of neuronal hyperactivity. However, the PLX treatment increased the plasma NfL levels in male wildtype mice but did not upregulated the expression of IEGs. Therefore, the expression changes in IEGs may not necessarily be correlated with the excitotoxicity.
4. In regard of intermittent CSF1R inhibition, how to exclude the complicated effects caused by microglial repopulation? It has been reported that microglia rapidly repopulate the entire brain upon removal of the inhibitor.
5. The study showed that the CSF1R inhibitor preferentially eliminates reactive microglia around tau deposits. Therefore, it's not surprised to observe that the CSF1R inhibition shifts gene expression patterns in Tg2541 mice towards wildtype. The authors should discuss any potential mechanisms underlying the increased sensitivity of tau-associated microglia to PLX.

AUTHORS REMARKS

We greatly appreciate the time and effort of the reviewers to provide excellent feedback on the first version of our manuscript. We noted several common themes amongst the three reviews which we have made a substantial effort to address by performing additional experiments, providing additional data analyses, and adding new discussion to the main text. Specifically, there was a common request for more systematic evaluation of the role that sex plays in microglial depletion and tau reduction by CSF1R inhibitors, which we have addressed by performing principal component analyses (Fig. 1n and Supplementary Fig. 1q) and extensive statistical analyses (Supplementary Data File 1). Given that microglial or tau reduction are not differentially impacted, we pursued our hypothesis that CSF1R inhibitors cause neuronal excitotoxicity in male mice, evidenced by dose-dependent immediate early gene (IEG) upregulation only in male Tg2541 mice treated with PLX5622 (Fig. 7h-j). We have now confirmed this result in a completely independent study using PLX3397 (Supplementary Fig. 12), demonstrating that this is a robust sex-specific effect of two different pharmacological CSF1R inhibitors. Multiple reviewers asked about the impact of CSF1R inhibition in the periphery, which we have addressed by further evaluation of a non-brain penetrant CSF1R inhibitor analog, and by measurements of liver toxicity (Fig. 6 and Supplementary Fig. 5). Finally, we were asked to validate the reduced astrogliosis with CSF1R inhibition, which we did by evaluating the neurotoxic A1 astrocytic gene signature (Fig. 8c,d), and by confirming our bioluminescence imaging (BLI) results with IHC and mRNA analysis of GFAP (Supplementary Fig. 13). Together, these new data add to our systematic approach which has improved our mechanistic understanding of the sex-dependent effects of CSF1R inhibitors, and our revised manuscript is now greatly improved. We discuss these new findings at length below, as well as other new data and discussion we have added in response to each reviewer comment.

REVIEWER COMMENTS

1. Reviewer #1:

1.1. In this manuscript, Johnson et al investigate the impact of CSF1R inhibition-mediated microglia depletion in a tauopathy model. This is of importance as tauopathy is considered to contribute to neurodegeneration, for instance in the common neurodegenerative disorder Alzheimer's disease, yet we do not have approved therapies to intervene. CSF1R inhibition is a commonly used tool in experimental models to deplete macrophages, and inhibitors are being trialled in humans for treatment of glioma and Alzheimer's disease. However we still do not understand the differential impact of different CSF1R inhibitors, treatment regimes, or sex on outcomes. Here, the authors administer CSF1R inhibitors to a mouse model of tauopathy and make 3 major findings. First, there are differential impacts of CSF1R inhibition on reducing tau pathology depending on the brain region/severity of pathology, and treatment paradigm. Second, there are sex differences in impacts on extending survival (females) or proxies of neurodegeneration (males). Third, CSF1R inhibition doesn't eliminate all microglia, indicating there may be resistant vs susceptible populations in the context of tauopathy. The novelty is in comparison of treatment paradigms and sexes, building on previous work that administered

CSF1R inhibitors to models of tauopathy only the male sex or investigated a single time point post treatment. With regards to the data, I have the following suggestions.

We thank the reviewer for their constructive feedback and general enthusiasm for our study. We addressed as many of your concerns and questions as possible to the best of our abilities.

Reviewer #1 Major Suggestions:

1.2. I appreciate that the authors excluded impacts of the CSF1R inhibitors on peripheral macrophages (including monocytes which could then travel to the brain). However there are other CNS-resident macrophages (border associated macrophages), and these are known to be depleted by CSF1R. Of particular interest are the perivascular macrophages, which are directly adjacent to brain parenchyma. Can the authors assess the impact of their depletion paradigms on perivascular macrophage numbers (e.g. Lyve1+ IBA1+ cells, in association with blood vessels).

Thank you, this is an interesting comment. Indeed there are other CNS-resident macrophages that do become depleted with CSF1R inhibitors. While we do not diminish the importance of these perivascular macrophages, their role in tauopathy is even more unclear. Generally these are understudied cells. But, as it stands, parenchymal microglia have a seemingly subtle morphological response to intraneuronal tau inclusions in mice, and thus it is not clear what we would learn in this instance by examining the mere distribution of Lyve1+ Iba1+ cells, nor would it change the major conclusions of this study. For the time being, performing this experiment is beyond the scope of this study. If in fact we were studying a mouse model of amyloid-beta amyloidosis with cerebral amyloid angiopathy it would be highly pertinent as there is likely a direct interaction between CAA on the blood vessel walls and adjacent PVMs. One could imagine a major independent study on this topic alone.

1.3. There are interesting suggestions throughout the paper suggesting impact on or rescue of neuronal damage – with NFL measured in the CSF, and the possibility of excitotoxicity is mentioned. As tauopathy is considered to be associated with neurodegeneration, can the authors assess neurodegeneration directly in the brain. Are there differences in the survival of neurons in the conditions where differences were observed, e.g. females with extended lifespans in PLX treated vs vehicle, males showing ‘toxic’ responses to high PLX blood/brain concentrations?

Tg2541 mice do exhibit nonapoptotic neurodegeneration but it is mostly limited to motor neurons in the anterior horn of the spinal cord (Allen et al. 2002, J Neurosci) with minimal frank neuronal loss in the cerebral cortex (Hampton et al. 2010, J Neurosci) and none in the hippocampus (Xu et al. 2014, Neuropathol Appl Neurobiol). Therefore, we found that NeuN and total tau immunoreactivity in forebrain and hindbrain structures were largely unchanged by CSF1R inhibition, with the exception of modestly increased NeuN staining following acute treatment with PLX5622 (Supplementary Fig. 6d). Our results are consistent with a prior report showing that PLX3397 does not affect neuronal viability or tau expression (Shi et al. 2019, J Exp Med), which has been referenced and discussed in the text (Line #187-194 and 514-521).

Thus, the functional deficits that we observe in Tg2541 mice that are rescued by CSF1R inhibition are likely caused by neuronal dysfunction (e.g., excitotoxicity) rather than neuronal death. Indeed, we found increased brain expression of immediate early genes (IEGs) (Fig. 7h-j and Supplementary Fig. 12) and increased plasma neurofilament light chain (NfL) (Fig. 5) in male, but not female, Tg2541 mice. Furthermore, we found that NfL levels measured in brain lysates from both forebrain and hindbrain correlate with PLX3397 concentration in male mice but not female mice, which has been added as Fig. 5i,j. Our results are consistent with several published studies that have separately linked tau accumulation or microglial modulation with neuronal excitotoxicity, as discussed in the text (Line # 525-537), yet our data uncover a sex-dependent response to CSF1R inhibition with concurrent tauopathy that was not previously known.

1.4 The two CSF1R inhibitors 5622 and 3397 are used interchangeably throughout the paper (i.e. both drugs are not used in every treatment paradigm), can the authors clarify why this is? It would be useful to at least have the quantification of microglia depletion for both drugs (as only 5622 is provided in the SFig1 but not for 3977) for IBA1 and P2RY12 measures, to be confident of experiments throughout the manuscript using 3977.

In Supplementary Fig. 1, we provide quantification of microglial depletion for both PLX3397 and PLX5622 by both Iba1 and P2RY12 immunohistochemical analysis, in both Tg2541 and wild type mice. We tested both PLX3397 and PLX5622 in some experiments in order to confirm the effects of CSF1R inhibition on tau using two different drugs that target the same pathway. We did not use both drugs in every experiment because it would have required an exorbitant amount of mice and cost. We will note, however, that both drugs were used at dosages recommended by Plexxikon Inc., and both drugs have been used in prior studies by other labs (e.g., Shi et al. 2019, J Exp Med; Bennett et al. 2018, J Neuroinflamm; Asai et al. 2015, Nat Neurosci; Dagher et al. 2015, J Neuroinflamm; Spangenberg et al. 2016, Brain; Son et al. 2020, Int J Mol Sci; Pinto et al. 2020, Neuron).

1.5. Given the focus on sex differences in the manuscript, it would be useful to see statistical comparisons between males and females for the 3 measures of tauopathy (as currently they are combined in Fig1).

We agree completely and we have assembled Supplementary Data File 1 containing a comprehensive statistical analysis of the effect of sex in every experiment where male and female mice are combined (40 figure panels total). Specifically, we evaluated whether male and female mice were different by testing the main effect of sex overall, we evaluated whether drug efficacy was dependent on sex by testing the interaction between sex and drug effect, and we evaluated the efficacy of each drug compared to vehicle within each sex by multiple comparisons testing. From this data, it is clear that the endogenous levels of tau and, at some ages, microglia are different between male and female mice, indicated by the main effect of sex in the 3-way ANOVA (Supplementary Data File 1a). However, we did not identify any meaningful sex*drug interaction effects by 3-way ANOVA, suggesting that the drugs are equally effective in male and female mice at depleting microglia and reducing pathogenic tau levels,

which is also confirmed by testing the main effect of drug in 2-way ANOVA analyses of male and female mice separately (Supplementary Data File 1b,c).

In order to quickly convey the overall effects of CSF1R inhibitors across the numerous experiments, we also performed principal component analysis. We first standardized the data from drug-treated mice to the respective vehicle-treated mice within sex and dosing paradigm groups, and then used both forebrain and hindbrain tau-prion cell bioassay and pS396 tau ELISA measurements to calculate a combined 'tau score', and both forebrain and hindbrain Iba1 and P2YR12 measurements to calculate a combined 'microglial score'. We identified two principal components (PC1 and PC2) for each model which accounted for 70.7% of the total variance in the tau model and 79.0% in the microglial model. We then used multiple linear regression to evaluate the main effects of sex and dosing paradigm, and the sex*dosing interaction effect of the drug-treated groups only. We also used multiple linear regression to evaluate the main effect of drug and the sex*drug interaction effect with the vehicle-treated groups included. These data are now presented in Fig. 1n (Tau score) and Supplementary Fig. 1q (Microglial score). Our principal component analyses confirm our interpretation of Supplementary Data File 1 described above, showing that both PLX3397 and PLX5622 reduce microglial score and tau score, and that sex does not influence the effect of either drug relative to vehicle. Therefore, we attribute the lack of survival extension in male mice to drug-induced neuronal excitotoxicity, as described in the manuscript (Line #329-354 and 525-537), rather than a lack of microglial or tau reduction.

1.6. Throughout the manuscript, it is not clear what the 'n' numbers in the graphs represent. Is every dot an individual mouse? Can this be clarified in the figure legend?

Throughout the manuscript, each symbol represents an individual mouse. This has been added to figure legends where it was missing.

1.7. Can the authors compare the gene expression differences from the nanostring between males and females, females with longer lifespan from PLX vs vehicle, and males with increased toxicity vs not? This could give indications as to why there are functional differences with the different groups following microglia depletion.

Indeed, at the beginning of this project we carefully considered the timing of each endpoint tissue collection for use in a specific downstream assay(s). Over several years of using the Tg2541 model, we observed that the median lifespan of these transgenic lines fluctuates from 7 to 8 months of age. After the first sign of gross motor deficit is noted (~5 months of age), the accumulation of additional neurological (motor) signs progresses in a gradual, predictable manner over 6 to 8 weeks. However, in the last stage of clinical disease, the animal declines rapidly and accumulates the final neurological signs at a rapid and less predictable rate, which ultimately require humane euthanasia. This final period can vary from mouse to mouse even if the onset of disease is somewhat synchronized. We suspected that this last phase of the severe clinical phenotype (e.g. hindlimb paralysis, failure to rear and eat, and consequent weight loss) may be associated with a higher degree of variability in Nanostring analysis given the acuteness of gene expression changes that can occur with rapid changes in general health. Thus, we

reasoned that to obtain the most robust data with sufficient biological replicates from both sexes in all treatment arms during the symptomatic phase of disease, we collected tissues at 7 months of age. This age represents a near end-of-life age and severe disease stage but allows us to time-lock the tissue collection to avoid the likely variations in gene expression due to mice perishing at different ages and experiencing different end of life behaviors due to differences in locomotion and feeding. Moreover, the RNA expression changes at 7 months of age likely represent the chronic and stable changes in gene expression associated with the protracted accumulation of tau pathology and gliosis and protection with PLX drugs. We are confident the current data set is in fact informing on the brain-wide changes associated with functional rescue and extended survival. These points are now discussed on Line #551-593 in the main text.

1.8. Is there statistical significance between the male vs female groups for the IEGs?

We have added additional statistical tests as requested. Indeed, male Tg2541 mice treated with PLX5622 showed a significantly higher expression level of IEGs compared to female PLX5622-treated Tg2541 mice ($P = 0.017$) and we have added this analysis to Fig. 7j. Importantly, we have also now confirmed this same effect in PLX3397-treated mice ($P = 0.039$, Supplementary Fig. 12).

1.9. Given the focus on the IEGs as a potential mechanism mediating differential responses to CSF1R, it would be useful to see validation of this at the qPCR or protein level. Ideally this would be on-tissue validation as induction of IEGs can occur in cells while tissue is being processed for sequencing.

We thank the reviewer for the suggestion. Indeed, we validated 53 genes by qPCR and found a strong correlation between mRNA levels using this method and Nanostring (Supplementary Fig. 9). Furthermore, we have directly evaluated *CSF1* and *CSF1R* gene expression and we found no sex-dependent differences in either wild type or Tg2541 mice treated with vehicle or with CSF1R inhibitors, and we have now included this data as Supplementary Fig. 11. Importantly, we have also now confirmed the effect of PLX5622 on dose-dependent upregulation of IEGs only in male Tg2541 mice (Fig. 7h-j) in a completely independent study using PLX3397 (Supplementary Fig. 12), demonstrating that this is a robust sex-specific effect of two different pharmacological CSF1R inhibitors.

1.10. Can the changes in astrocyte reactivity indicated by bioluminescence be validated by staining for GFAP in the tissue?

This is an excellent suggestion and we have indeed validated our bioluminescence imaging (BLI) results by GFAP immunostaining in brain sections and have added that new data, along with representative images, as Supplementary Fig. 13. As shown, PLX3397 treatment reduces the percent area of GFAP+ immunolabeling in both forebrain and hindbrain regions in the brains of Tg2541 mice, consistent with the reduction in astrocyte reactivity by PLX3397 observed using BLI (Fig. 8f,g). Furthermore, we have also measured *GFAP* mRNA expression in brain samples and have added that data as Supplementary Fig. 13c,d, which shows that *GFAP* expression is significantly reduced in the hindbrain of PLX-treated mice compared to vehicle-treated mice.

Additionally, we comprehensively evaluated astrocytic genes in both brain regions and we observed reductions in the detrimental astrocytic A1 gene cluster with PLX treatment, now included as Fig. 8c,d.

1.11. The authors suggest that the PLX drug toxicity in the male mice is due to off-target effects independent of microglia e.g. anemia, hepatotoxicity, but it wasn't clear to me that this is necessarily the case. Are there differences in neurodegeneration/neuronal health that are the cause? Do the mice actually show hepatotoxicity or anemia? It's quite important to know what underpins the deleterious responses in males for translational relevance. Comparing the males that did poorly to the ones on the intermittent diet (where they did better) could be interesting.

We thank the reviewer for bringing up this important point. Our assumption that PLX3397 may have off-target effects was based on a previous report using the drug in a different mouse model (Shi et al. 2019, J Exp Med). As noted in the text, PLX3397 (pexadartinib) has also been shown in numerous clinical trials to result in adverse effects including anemia, leukopenia, and hepatotoxicity (Benner et al. 2020, Drug Des Devel Ther). Furthermore, it has been shown that male mice may be more sensitive to microglial depletion than female mice (Berve et al. 2020, J Neuroinflammation).

As such, we performed histopathological analyses including H&E, Masson's trichrome, and Picosirius red staining of fibrosis in liver sections of Tg2541 mice as an indicator of hepatotoxicity, and we have added that data as Fig. 6i-k. However, we did not observe a significant difference between PLX3397- and vehicle-treated mice, or between male and female Tg2541 mice. As a more sensitive method, we also analyzed plasma for levels of alkaline phosphatase (ALP), another well-accepted measure of liver damage. We found blood ALP levels to be elevated in both male and female Tg2541 mice following treatment with PLX3397, and we have added that data as Fig. 6l. These results indicate that the sex-specific drug toxicity in Tg2541 mice likely occurs in the brain (e.g. neuronal excitotoxicity), rather than as hepatotoxicity, as we now discuss in the text (Line # 275-294). As the reviewer suggests, male mice receiving intermittent dosing of PLX3397 did indeed have reduced levels of plasma neurofilament light chain (NfL) (Fig. 5m), one indicator of neurotoxicity, whereas continuous dosing led to increased plasma NfL (Fig. 5c,f,p). In new data, we validate NfL as a marker of drug-induced neurotoxicity by showing that NfL levels measured in brain lysates from both forebrain and hindbrain correlate with PLX3397 concentration in male mice but not female mice, which has been added as Fig. 5i,j. Lastly, in new experiments we found that PLX73086, a non-brain penetrant CSF1R inhibitor analog, does not increase NfL levels in either Tg2541 or wild type mice (Fig. 6a-h), indicating that drug-induced NfL release is dependent on the drug entering the brain rather than only peripheral exposure.

Reviewer #1 Minor suggestions:

1.12. Can the authors discuss why there are some different responses seen between 3397 and 5622 with regards to impact on the 3 measures of tau pathology? Can the authors also discuss the implications of not all 3 measures being impacted in the same way even with the same drug and treatment regimen?

We tested two CSF1R inhibitors (PLX3397 and PLX5622) in three treatment regimens (acute, chronic, and terminal), and we evaluated tau pathology by three measures (Tau-prion bioassay, ELISA, and IHC) in two brain regions (forebrain and hindbrain). Although we did not observe a reduction in all three measures by both inhibitors in all three treatment regimens in both brain regions, we observed a clear trend of reduced tau pathology in Tg2541 mice caused by CSF1R inhibition (Fig. 1). Since the various measures of tau pathology represent different steps of tau pathogenesis (hyperphosphorylation vs. oligomerization vs. filament formation), they may be differentially impacted by CSF1R inhibition with different treatment regimens. However, our principal component analysis clearly shows that, overall, both PLX3397 and PLX5622 reduce tau score in both male and female mice. These data have been added as Fig. 1n and additional discussion of this topic can now be found in Line #120-144.

1.13. In supplemental figure 8, it looks like in the K18 fibril model the PLX treated males do indeed have an extended lifespan, which counters the sex specific conclusions from the paper. Can the authors clarify?

The figure (now Fig. 3h) shows that there is not a significant effect of PLX3397 treatment on lifespan of male mice inoculated with K18 fibrils compared to vehicle treatment ($P = 0.1205$).

1.14. As the authors mentioned that microglia inflammasome activation can regulate tauopathy, can the authors check if inflammasome related genes were changed in the nanostring analysis with PLX treatment?

Following the reviewer's recommendation, we examined 14 inflammasome-related genes in our dataset. At least in the Tg2541 mouse model, we did not observe obvious activation in these genes, suggesting that tau pathology triggers a specific activation program in the microglia cells that is different from the inflammation phenotype that may occur in more aggressive transgenic tau models or due to different background strains (Neuner et al. 2019, Neuron). These data are now included in Supplementary Fig. 17 and this topic is discussed on Line# 397-426.

1.15. Can the authors change the title of figure 7 to more accurately explain what was done, as the current title referring to 'tau resistant' microglia is confusing and specific microglia subpopulations were not sorted then sequenced. Referring to wild type vs transgenic bulk gene expression more accurately reflects what was done.

We thank the reviewer for the suggestion and agree that the subpopulation refers to our interpretation of the data. We have now changed the title of the figure (now Fig. 10) to "Selective ablation of tau-activated microglia gene expression by PLX5622".

1.16. Can the authors include representative images of microglia depletion with PLX with respect to proximity to tau deposits (currently only the vehicle control images are shown).

We have added new representative confocal images of microglia with and without PLX treatment to Fig. 9.

1.17. Can the authors discuss – in the discussion – the implications for differences seen between forebrain and hindbrain responses in their model? What are the implications of human disease (regional responses, responses based on severity of tauopathy, etc).

We previously identified a regional vulnerability to tau prion propagation in the brains of Tg2541 mice, causing earlier and greater tau accumulation in hindbrain regions compared to forebrain regions (Johnson et al. 2017, Proc Nat Acad Sci USA). This is consistent with the staging of the human primary tauopathy progressive supranuclear palsy (PSP), where tau deposition begins and predominates in subcortical and brainstem nuclei (Kovacs et al. 2020, Acta Neuropathol), now discussed on Line #107-112. We observed greater efficacy of early (acute) CSF1R inhibition to restrict tau-prion levels in forebrain regions (Fig. 1e), likely due to reduced disease severity there relative to hindbrain regions. At later stages (chronic and terminal), CSF1R inhibition with PLX3397 did reduce tau-prion levels in the hindbrain, albeit a modest reduction relative to the effects in the forebrain (Fig. 1f,g). These findings suggest that early and long-term CSF1R inhibition (though not necessarily continuous) would most effectively mitigate human tauopathy, as we have now discussed in the text (Line # 551-593).

1.18. The authors refer to DAMs as being known to have a pro inflammatory profile and be detrimental, this is not correct and needs to be amended. The DAM profile has been linked with anti inflammatory microglia profiles and even developmental and regenerative processes.

We have corrected the language in the text as suggested on Line # 392-394.

2. Reviewer #2

2.1. The manuscript by Johnson et al. describes an interesting concept, connecting microglia and tauopathy and gender. Yet, the manuscript in its present form reads like a collection of partial stories that were not fully explored or integrated, and the conclusions are appealing but not fully supported and mechanistically clear.

We appreciate the supportive comments and we have addressed the reviewer's concerns and questions to the best of our abilities. We believe the manuscript is now greatly improved.

We acknowledge the reviewer's point-of-view regarding a lack of integration in the story line; however, we would argue that the common thread throughout the entire manuscript is a detailed evaluation of sex, drug pharmacokinetics, and disease phenotypes. Beyond interesting, sex as a biological variable is a crucial component of preclinical studies and understanding how sex dictates disease progression or drug response is vital for the success of putative therapeutics. Contrary to the recent works in the field, we have assembled a massive study in which we run multiple parallel experimental arms to study multiple time points, treatment paradigms, end-point measurements and longitudinal biomarkers with clinical relevance. Too often the field of preclinical research is left trying to piece together results from different labs using different models, conditions, end-point assays and drugs for the same target. Major leaps of faith must be taken to decipher what can be reasonably compared and extrapolated. All the while one has

to accept the caveats of what the authors of a given publication show and don't show the reader (negative data) as a matter of convenience for story-telling. Considering that the A β immunotherapy approach for AD has been in development for 3 decades and there is only just now a hint of success in clinical trial data, we would argue that this leaves the door wide open for much more preclinical research on CSF1R inhibitors in neurodegenerative disease models. This class of drugs is already in a human trial for AD, but is based on limited preclinical data and, importantly, none of those publications considered sex as a variable.

2.2. Figure 1: The authors found that depletion of microglia had a different outcome on pathology, in different parts of the brain. Moreover, assessment by ELISA and by immunohistochemistry gave a different picture. Genders of the mice should be presented separately. In addition, it is puzzling why the early and brief depletion is more effective than continuous depletion, as evaluated by IHC.

We agree completely and we have assembled Supplementary Data File 1 containing a comprehensive statistical analysis of the effect of sex in every experiment where male and female mice are combined (40 figure panels total). Specifically, we evaluated whether male and female mice were different by testing the main effect of sex overall, we evaluated whether drug efficacy was dependent on sex by testing the interaction between sex and drug effect, and we evaluated the efficacy of each drug compared to vehicle within each sex by multiple comparisons testing. From this data, it is clear that the endogenous levels of tau and, at some ages, microglia are different between male and female mice, indicated by the main effect of sex in the 3-way ANOVA (Supplementary Data File 1a). However, we did not identify any meaningful sex*drug interaction effects by 3-way ANOVA, suggesting that the drugs are equally effective in male and female mice at depleting microglia and reducing pathogenic tau levels, which is also confirmed by testing the main effect of drug in 2-way ANOVA analyses of male and female mice separately (Supplementary Data File 1b,c).

In order to quickly convey the overall effects of CSF1R inhibitors across the numerous experiments, we also performed principal component analysis. We first standardized the data from drug-treated mice to the respective vehicle-treated mice within sex and dosing paradigm groups, and then used both forebrain and hindbrain tau-prion cell bioassay and pS396 tau ELISA measurements to calculate a combined 'tau score', and both forebrain and hindbrain Iba1 and P2YR12 measurements to calculate a combined 'microglial score'. We identified two principal components (PC1 and PC2) for each model which accounted for 70.7% of the total variance in the tau model and 79.0% in the microglial model. We then used multiple linear regression to evaluate the main effects of sex and dosing paradigm, and the sex*dosing interaction effect of the drug-treated groups only. We also used multiple linear regression to evaluate the main effect of drug and the sex*drug interaction effect with the vehicle-treated groups included. These data are now presented in Fig. 1n (Tau score) and Supplementary Fig. 1q (Microglial score). Our principal component analyses confirm our interpretation of Supplementary Data File 1 described above, showing that both PLX3397 and PLX5622 reduce microglial score and tau score, and that sex does not influence the effect of either drug relative to vehicle. Therefore, we attribute the lack of survival extension in male mice to drug-induced

neuronal excitotoxicity, as described in the manuscript (Line #329-354 and 525-537) rather than a lack of microglial or tau reduction.

2.3. This experiment is missing visualization of the extent of depletion, and to what extent homing of myeloid cells from the periphery took place.

In Supplementary Fig. 1, we provide quantification of microglial depletion for both PLX5622 and PLX3397 by both Iba1 and P2RY12 immunohistochemical analysis, in both Tg2541 and wild type mice. We also state in the main text: “in our study, neuroprotection occurred despite incomplete microglia depletion (~60%)”, which indicates the average percent microglial depletion across all experiments. To provide a visualization of the extent of depletion, we have added new representative images as Fig. 9a showing the level of microglial depletion by IHC staining in brain sections of Tg2541 mice treated with vehicle or PLX3397.

To the reviewer’s point about homing of peripheral myeloid cells to the CNS. This had been a controversial topic for some time. There is a body of literature arguing that infiltration of peripheral myeloid cells occurs in neurological disease states, but these data were generated using an irradiation-based bone marrow replacement paradigm, which is (as we now know) fraught with many caveats even when using the head-shielded permutation. This irradiation approach, and the data generated from it have now been refuted by many labs that use the parabiosis model to study infiltrating cells in brain health and disease. It has been shown that peripheral myeloid cells do not enter the adult brain to any appreciable degree in several models of neurodegenerative disease (Ajami et al. 2007, *Nat Neurosci*; Wang et al. 2016, *J Exp Med*). Although homing of peripheral myeloid cells may occur in some pathological and brain injury conditions with overt loss of blood-brain barrier integrity (such as stroke or TBI), the numerous parabiosis experiments have demonstrated that blood-derived myeloid cells are minimal to nonexistent in the central nervous system in the context of neurodegenerative disease models lacking major BBB deficits (such as our tauopathy model), which instead attains myeloid cells directly from the skull and vertebral bone marrow (Herisson et al. 2018, *Nat Neurosci*; Cugurra et al. 2021, *Science*; Brioschi et al. 2021, *Science*). Even in the case where minute numbers of peripheral myeloid cells may enter the brain, their contribution is expected to be marginal at best considering the proportion of brain-resident microglia. Furthermore, we have now ruled out any robust role of peripheral myeloid cells by using a non-brain penetrant CSF1R analog (PLX73086) to deplete only peripheral cells and show that it does not deplete microglia or reduce tau deposition (Supplementary Fig. 5), or affect plasma NfL levels or body weight in either wild type or Tg2541 mice (Fig. 6a-h).

2.4. Figure 2: The authors describe gender differences without any mechanistic explanation. Moreover, the presentation of the entire figure is confusing.

We did provide a mechanistic explanation for the sex-specific effects of PLX3397 on survival of Tg2541 mice (Fig. 3b,h) in the Results sections. Briefly, we first showed that tau-prion levels were correlated with survival in female mice, but not male mice (Fig. 3e), suggesting that tau-prions are the main cause of death in female mice. Since we showed definitively that tau-prions were reduced by PLX3397 in both male and female mice (Fig 1 and Supplementary Data File 1), we postulated that drug metabolism or toxicity may also be involved. We then showed that

female mice are hyperactive (Fig. 4e-g) indicative of a higher metabolism than in male mice. Consistent with this premise, we also found that male mice had higher drug concentrations in plasma (Fig. 4b) and increased NfL levels that were correlated with decreased survival (Fig. 5g,h), indicative of drug toxicity in male mice. These data, taken together with extensive supplementary data, suggest that despite similar dosing levels, female mice have a lower PLX3397 exposure which reduces tauopathy and increases survival, while male mice have a higher PLX3397 exposure leading to neuronal excitotoxicity and prevents an extension of survival, as detailed in the Results and Discussion. In order to make the figures easier to follow, we have added subtitles to each panel in the main figures.

To provide additional mechanistic insight, we identified sex-dependent neuronal excitotoxicity as indicated by increased brain expression of immediate early genes (IEGs) (Fig. 7h-j) and increased plasma neurofilament light chain (NfL) in male, but not female, Tg2541 mice (Fig. 5b,c,e,f). Importantly, we have also now confirmed the effect of PLX5622 on dose-dependent upregulation of IEGs only in male Tg2541 mice in a completely independent study using PLX3397 (Supplementary Fig. 12), demonstrating that this is a robust sex-specific effect of two different pharmacological CSF1R inhibitors. Furthermore, we found that NfL levels measured in brain lysates from both forebrain and hindbrain correlate with PLX3397 concentration in male mice but not female mice, which has been added as Fig. 5i,j. Our results are consistent with several published studies that have separately linked tau accumulation or microglial modulation with neuronal excitotoxicity, as discussed in the text (Line # 525-537), yet our data uncover a sex-dependent response to CSF1R inhibition with concurrent tauopathy that was not previously known.

2.5. Figure 3: Not all correlations with NFL shown in the figure add to the story.

We evaluated blood plasma neurofilament light (NfL), a validated biomarker of neuronal injury, to interrogate potential mechanisms of sex-specific PLX efficacy. As shown in Fig. 5c,f,p and described in the Results section, PLX treatment increased NfL levels in both male Tg2541 mice and in male wild type mice, indicative of drug toxicity, and in Tg2541 mice NfL levels were correlated with decreased survival (Fig. 5h). However, PLX treatment reduced NfL levels in female Tg2541 mice (Fig. 5b) and NfL levels were not correlated with survival (Fig. 5g) suggesting that PLX had minimal/no toxic effects in female mice. In new data, we also validated NfL as a marker of neurotoxicity by showing that NfL levels measured in brain lysates from both forebrain and hindbrain correlate with PLX3397 concentration in male mice but not female mice, which has been added as Fig. 5i,j. Finally, our new experiments show that a non-brain penetrant CSF1R analog (PLX73086) does not affect plasma NfL levels or body weight in either wild type or Tg2541 mice (Fig. 6a-h), indicating that NfL as a readout is indeed CNS-dependent.

2.6. Figure 5: Gliosis should be measured by IHC.

This is an excellent suggestion and we have indeed validated our bioluminescence imaging (BLI) results by GFAP immunostaining in brain sections and have added that new data, along with representative images, as Supplementary Fig. 13. As shown, PLX3397 treatment reduces the percent area of GFAP+ immunolabeling in both forebrain and hindbrain regions in the brains of Tg2541 mice, consistent with the reduction in astrocyte reactivity by PLX3397 observed using BLI (Fig. 8f,g). Furthermore, we have also measured *GFAP* mRNA expression in brain samples and have added that data as Supplementary Fig. 13c,d, which shows that *GFAP* expression is

significantly reduced in the hindbrain of PLX-treated mice compared to vehicle-treated mice. Additionally, we comprehensively evaluated astrocytic genes in both brain regions and we observed reductions in the detrimental astrocytic A1 gene cluster with PLX treatment, now included as Fig. 8c,d.

2.7. Figure 6: IB1 is not a valid marker for microglia, certainly not in a case in which microglia are partially depleted.

We validated our Iba1 immunohistochemical analysis using a second pan-microglial marker, P2RY12, as presented in Supplementary Fig. 1, and the results are largely consistent. Iba1 has been used as a microglial marker in nearly every study involving microglial depletion via CSF1R inhibition (eg. Spangenberg et al. 2019, Nat Commun; Szalay et al. 2016, Nat Commun; Asai et al. 2015, Nat Neurosci; Bennett et al. 2018, J Neuroinflamm; Dagner et al. 2015, J Neuroinflamm; Elmore et al. 2018, Aging Cell; Henry et al. 2020, J Neurosci; Mancuso et al. 2019, Brain; Pinto et al. 2020, Neuron; Seitz et al. 2018, J Virol; Shi et al. 2019, J Exp Med; Son et al. 2020, Int J Mol Sci; Sosna et al. 2018, Mol Neurodegen; Spangenberg et al. 2016, Brain). In keeping consistent with this standard in the field, we were able to directly compare the levels of microglial depletion seen in our experiments with those seen in prior studies, as discussed at length in the text (Line #429-461).

3. Reviewer #3

3.1. The current study use a comprehensive treatment paradigms of CSF1R inhibitors to evaluate the roles of CSF1R inhibition in Tau-related pathology. They observed consistent reduction in the levels of pathogenic tau in the brains of both male and female Tg2541 mice with PLX treatment. Interestingly, only female mice benefited from extended survival, functional rescue, and reduced NfL levels. Hence, the authors raise an important issue to carefully evaluate the role of sex in the translational studies that target CSF1R inhibition. Overall the paper is descriptive and provides incremental knowledge to the field.

We thank the reviewer for the constructive comments. We have worked hard to address the reviewer's concerns, and we believe the manuscript is now greatly improved. However, we are surprised by the reviewer's opinion that our study provides only incremental knowledge to the field. No prior study in the field has reported on sex as a major variable in the functional recovery from CSF1R inhibition in a model of neurodegenerative disease. Moreover, no prior study has considered that complete microglia ablation may be unnecessary or may even be counter-productive. Our study presents numerous novel findings that have significant implications for basic science and for clinical translation of CSF1R inhibitors and other neuroimmune modulators.

3.2. The sex-specific differences could be attributed to different plasma and brain levels of PLX3397 between male and female mice. However, it remains unclear why the different drug levels did not impact its effects on reduction of microglia number and tau pathology.

Indeed, we observed sex-specific differences in the levels of PLX3397 in the plasma and brains of both Tg2541 and wild type mice (Fig. 4b and Supplementary Fig. 7). As detailed in the main text, we do attribute sex-specific differences in PLX3397 efficacy to these differences in drug levels, as evidenced by Fig. 4c showing that there is a direct correlation between plasma PLX3397 levels and microglial number in both the forebrain and hindbrain. Furthermore, Fig. 4d shows there is a strong correlation between plasma PLX levels and tau-prion activity in the forebrain. Thus, different drug levels do dictate the reduction of microglia and tau pathology within groups of male and female Tg2541 mice. However, we identified sex-dependent drug-induced neuronal excitotoxicity as indicated by increased brain expression of immediate early genes (IEGs) (Fig. 7h-j and Supplementary Fig. 12) and increased plasma neurofilament light chain (NfL) in male, but not female, Tg2541 mice (Fig. 5b,c,e,f). Furthermore, we found that NfL levels measured in brain lysates from both forebrain and hindbrain correlate with PLX3397 concentration in male mice but not female mice, which has been added as Fig. 5i,j. Our results are consistent with several published studies that have separately linked tau accumulation or microglial modulation with neuronal excitotoxicity, as discussed in the text (Line # 525-537), yet our data uncover a sex-dependent response to CSF1R inhibition with concurrent tauopathy that was not previously known.

3.3. The study did not have data demonstrating why the male mice tend to accumulate higher amounts of CSF1R inhibitors in their plasma and brain. Is it due to slower metabolism of the drug?

To interrogate potential mechanisms of sex-specific levels of PLX in the plasma and brain, we performed automated home cage monitoring experiments. As shown in Supplementary Fig. 8, male and female mice had similar food consumption (drug intake) rates. However, male mice were consistently less active than female mice (in both PLX3397 and PLX5622 cohorts) which led us to propose that male mice have a slower overall metabolic rate and therefore metabolize PLX slower than female mice, as discussed in the text (Line #219-236).

3.4. The study showed higher expression of immediate early genes (IEGs) in male mice which may be indicative of neuronal hyperactivity. However, the PLX treatment increased the plasma NfL levels in male wild type mice but did not upregulate the expression of IEGs. Therefore, the expression changes in IEGs may not necessarily be correlated with the excitotoxicity.

PLX treatment may have multiple, yet to be identified, adverse effects in the CNS. For example, PLX may induce different mechanisms causing NfL increase in the plasma or upregulation of IEGs. One thing is clear—the adverse effects observed require that PLX3397 or PLX5622 enter the brain, as PLX73086 (non-brain penetrant analog) did not cause an increase in NfL in either Tg2541 or wild type mice (Fig. 6a-h). As the reviewer points out, IEGs were also not significantly upregulated in male WT mice, indicating that their activation may not be due to high concentration of PLX alone, but may also be dependent on tau deposits. Previous studies have linked tau accumulation and aberrant neural activity in vivo (Busche et al. 2019, Nat Neurosci). On the other hand, microglia are known to mediate neuroprotection against excitotoxicity and elimination of microglia can exacerbate seizures and related neuronal degeneration (Badimon et al. 2020, Nature; Liu et al. 2020, Am J Physiol Cell Physiol; Vinet et al. 2012, J

Neuroinflammation). Therefore, the concurrent tau removal and microglial elimination may increase the risk for hyperactivity, resulting in excitotoxicity in male mice with high PLX concentrations. Additional discussion of this topic has been added to the text on Line #521-537. Importantly, we have also now confirmed the effect of PLX5622 on dose-dependent upregulation of IEGs only in male Tg2541 mice (Fig. 7h-j) in a completely independent study using PLX3397 (Supplementary Fig. 12), demonstrating that this is a robust sex-specific effect of two different pharmacological CSF1R inhibitors.

3.5. In regard of intermittent CSF1R inhibition, how to exclude the complicated effects caused by microglial repopulation? It has been reported that microglia rapidly repopulate the entire brain upon removal of the inhibitor.

Thank you for raising this point. Yes there is rapid repopulation, and at the drug doses used in our study, complete repopulation of microglia occurs by 21 days after mice are taken off PLX-containing chow (Elmore et al. 2015, PLoS One; Najafi et al. 2018, Glia). According to these published reports, the repopulated microglia achieve baseline status in most morphological measurements and transcriptional changes by 21 days after drug withdrawal. Thus, we reasoned that 21 days was a good window for the intermittent treatment paradigm because we would re-initiate treatment at a time when the brain had baseline-like microglia. We have included additional text to elaborate on the rationale of our intermittent paradigm on Line #115-158. To be clear, this study is not about microglia depletion in binary terms. We apply a drug that causes CSF1R inhibition and only partial ablation of microglia cells--we are not making any claims about therapeutic benefits when 100% microglia are removed (and based on our data we believe that this may actually be detrimental!). No doubt there could be an added complexity, but in spite of this, the fact remains that the intermittent treatment led to reduced pathogenic tau comparable to the continuous treatment arm, and for male mice, was less toxic as evidenced by the lower plasma NfL levels (Fig. 5m).

3.6. The study showed that the CSF1R inhibitor preferentially eliminates reactive microglia around tau deposits. Therefore, it's not surprised to observe that the CSF1R inhibition shifts gene expression patterns in Tg2541 mice towards wild type. The authors should discuss any potential mechanisms underlying the increased sensitivity of tau-associated microglia to PLX.

Thank you for encouraging us to expand on the interpretations of this novel finding. Indeed, based on all the published data in transgenic APP mouse models where the PLX-resistant microglia are those that can be found clustered around amyloid plaques, it was surprising to us that we found the opposite result in a tauopathy model. Microglia in regions of high tau burden were the ones eliminated while in adjacent areas distal to tau-laden neurons microglia persisted. Moreover, the persistent microglia appeared relatively normal looking. We have provided additional discussion of the possible mechanistic reasons for increased sensitivity of tau-associated microglia to PLX on Line #481-504 and 521-537.

REVIEWER COMMENTS

Reviewer #1 (Remarks to the Author):

I appreciate the authors performing additional experiments and analyses to address the issues I raised. This includes assessing neuronal impact in the brain directly (which they did by measuring NFL levels in brain lysates), performing additional statistical analyses on sex differences, validating a large number of IEG gene changes using qPCR, confirming astrocyte reactivity changes by staining, and directly assessing hepatotoxicity. These have improved the paper. I also appreciate that this model requires long term experiments.

I have some remaining issues as described below:

- 1. With regards to my suggesting to assess perivascular macrophage depletion after CSF1R inhibition, I understand the authors' point that these are understudied cells so their contribution to the results are unclear. I disagree however, that determining whether they are depleted in their CSF1R treatment paradigms would not add to the paper and is beyond the scope of the study. CSF1R inhibitors are known to deplete perivascular macrophages. All of the interpretations made in the study relate to the depletion of 'microglia', however it's unlikely that perivascular macrophages weren't also depleted. Although we don't know their contribution to pathology or the effects observed in this study, knowing if and to what extent they are depleted is important in informing whether we need to take this into account in interpreting the data. I agree that assessing the role of perivascular macrophages to the results observed is beyond the scope of the paper, but understanding which macrophage populations are depleted in the CNS with CSF1R inhibition is very much relevant to the conclusions drawn here. Staining for perivascular macrophages is an established protocol.**
- 2. For the validation of the gene changes by qPCR, I appreciate the large number of genes the authors have assessed in Supplemental Fig.9. As I had requested validation of the IEG genes, it would be helpful to separate out the IEG gene validations from the rest to help assess whether the IEG changes reported in the Nanostring are validated by qPCR, including statistics between conditions for each gene, or statistics on the IEG gene set from qPCR. Is this in the male mice only? In addition, the results indicates that 'The RT-qPCR results matched the trends shown in the Nanostring data (Supplementary Fig. 9), indicating that our transcriptomic data was robust', but there are some differences seen with the Nanostring results as shown in 'c' particularly for the PLX vs Veh in the transgenic. Is it appropriate to state that the results matched? How does this influence the interpretation of the changes (for non validated genes)?**
- 3. I appreciate the authors' explanation that doing gene expression analysis at end of life is not practical and variable, and understand the end point of 7 months for the analysis. However this should not preclude the requested comparison of female vs male gene expression from the Nanostring dataset – as it is there are 2 genes highlighted showing sex differences in Fig.7f. Are these the only 2 genes which were significantly different between transgenic mice + PLX in males vs females?**
- 4. For the brain NFL measurements, this does show a nice correlation between increasing concentration of PLX and increased brain NFL in males, but not females (new Fig.5i,j). The female NFL levels actually appear higher than the male ones in the hindbrain. Can the authors explain how this could mean that the males treated with PLX have increased neurotoxicity vs females?**
- 5. With regards to the astrocyte signature, the field is moving away from using A1 and A2 terminology to avoid assumptions about function based on expression of a few genes. I suggest rephrasing to say 'genes previously associated with neurotoxic astrocytes following LPS exposure or neuroprotective function in an artery occlusion model', or something to that effect.**

I have been asked to comment on responses to Reviewer 3's issues as they are unable to re-review, and I've done this to the best of my ability below:

- 1. I disagree with Reviewer 3 that this paper is incremental as the impact of sex on differential response to CSF1R inhibition in pathology has not been reported to my knowledge, and is important to consider for the field not just for potential therapeutic applications but also for experimental use of CSF1R inhibitors in models of disease.**
- 2. I also disagree with the opinion on the inappropriate use of IBA1, as although this can pick up all macrophage populations in the CNS, the more microglia 'specific' markers with good antibodies available are downregulated in the context of chronic injury so are likely not going to work. This does come back to my issue on perivascular macrophage depletion though, so IBA1 will pick up presence and depletion of both populations, and this should be acknowledged in the paper.**
- 3. The reviewer asked whether there was an impact on metabolism in females vs males, and although the authors had previously implied differences in metabolism based on activity measurement, however metabolism was not directly measured. I can't comment on whether this is what the reviewer was getting at nor how this would best be measured. This is what I assumed from the question as the activity data was already included in the first submission.**
- 4. The reviewer raises the point that IEGs can go up with PLX treatment without concomitant increase in NFL. I accept the authors' answer that the presence or absence of tau likely influences the sensitivity of neurons to the PLX conditions.**
- 5. The reviewer brings up microglia repopulation. The authors mention that previous papers showed that microglia repopulate to normal baseline levels, and this is when they decided to reinitiate the treatment. I think the rationale makes sense, but one must acknowledge that microglia repopulation in a pathological setting does not always lead to microglia coming back to baseline transcriptional profiles (eg. in aging), so this may have happened in the transgenic as well. This should be acknowledged in the paper in the discussion.**

Reviewer #2 (Remarks to the Author):

The authors made impressive efforts to address the referee' comments. Yet, the manuscript still suffers from being "patchy" rather than one coherent scientific discovery. In addition, it still misses a mechanistic explanation for the sex differences that is important and interesting but requires a more comprehensive mechanistic insight.

AUTHORS REMARKS

We greatly appreciate the time and effort of the reviewers to provide excellent feedback on the second version of our manuscript. We are excited to provide new data, new analyses, and new text that solidify our mechanistic understanding of the sex-dependent effects of CSF1R inhibitors.

REVIEWER COMMENTS

1. Reviewer #1:

I appreciate the authors performing additional experiments and analyses to address the issues I raised. This includes assessing neuronal impact in the brain directly (which they did by measuring NFL levels in brain lysates), performing additional statistical analyses on sex differences, validating a large number of IEG gene changes using qPCR, confirming astrocyte reactivity changes by staining, and directly assessing hepatotoxicity. These have improved the paper. I also appreciate that this model requires long term experiments. I have some remaining issues as described below:

1.1. With regards to my suggesting to assess perivascular macrophage depletion after CSF1R inhibition, I understand the authors' point that these are understudied cells so their contribution to the results are unclear. I disagree however, that determining whether they are depleted in their CSF1R treatment paradigms would not add to the paper and is beyond the scope of the study. CSF1R inhibitors are known to deplete perivascular macrophages. All of the interpretations made in the study relate to the depletion of 'microglia', however it's unlikely that perivascular macrophages weren't also depleted. Although we don't know their contribution to pathology or the effects observed in this study, knowing if and to what extent they are depleted is important in informing whether we need to take this into account in interpreting the data. I agree that assessing the role of perivascular macrophages to the results observed is beyond the scope of the paper, but understanding which macrophage populations are depleted in the CNS with CSF1R inhibition is very much relevant to the conclusions drawn here. Staining for perivascular macrophages is an established protocol.

The reviewer is correct that there will be effects of CSF1R inhibition on PVMs and we should consider this when interpreting our results. Indeed, of all the other CNS-associated macrophages, it would seem that PVMs would be the most likely to contribute along with parenchymal microglia to tauopathy and be affected by PLX compounds in a way relevant to our study. To this end, we have evaluated PVM levels in a large number of both male and female PLX3397-treated mice. First, we surveyed several IHC-validated antibodies based on the literature and vendor specifications in a small subset of tissues. As the reviewer suggested, we tested three different antibodies for Lyve1 (Novus #NB100-725, Novus #NB600-1008, R&D Systems #AF2125), which is a published marker of PVMs. However, as seen in the figure below, in addition to staining PVMs adjacent to blood vessels, we observed some non-specific staining with all three antibodies in a pattern resembling NeuN labeling, which prevented accurate quantification. Using the best antibody of the three, we did try to further optimize the staining using different antigen retrieval methods and antibody concentrations, but could not achieve a high signal-to-noise labeling. This could just be related to how our formalin-

fixed/paraffin-embedded (FFPE) tissues were preserved and processed. Unfortunately, we did not have any non-FFPE tissues remaining to continue these tests. Therefore, we evaluated CD206 as an alternative marker of PVMs that has been validated to overlap with Lyve1⁺ cells both transcriptionally and by IHC, including in the brain (Utz et al. Cell, 2020). We found a CD206 antibody (Biorad #MCA2235) that yielded consistent staining of PVMs without the non-specific signals observed in Lyve1 staining (also shown in the image below), and representative confocal microscopy images and quantification of the numbers of Iba1+/CD206+ PVMs in cortical blood vessels of a large number of male and female Tg2541 are now included as Supplementary Fig. 7. We did not detect a significant difference in the numbers of PVMs in PLX- or vehicle-treated mice of either sex, although there was a trend ($P = 0.0947$) towards reduced PVMs in female mice. In our dosing regimens where only ~60% of brain microglia are depleted, it could be possible that PVMs are not depleted by the same proportion—perhaps PVMs have different tolerance to CSF1R inhibition than microglia. Nonetheless, we have now referenced the fact that PLX has been shown before to deplete PVMs in mice (Kerkofs et al. 2020, Theranostics) and that this could possibly have influenced our results. Our new results and discussion of PVMs are now included on Lines 180-198 and 496-502.

PVM labeling with Lyve1 (Novus #NB100-725) or CD206 (Biorad #MCA2235) and co-labeled with Iba1 (Abcam #ab178847). Scale bars = 100 μ m.

1.2. For the validation of the gene changes by qPCR, I appreciate the large number of genes the authors have assessed in Supplemental Fig. 9. As I had requested validation of the IEG genes, it would be helpful to separate out the IEG gene validations from the rest to help assess whether the IEG changes reported in the Nanostring are validated by qPCR, including statistics between conditions for each gene, or statistics on the IEG gene set from qPCR. Is this in the male mice only? In addition, the results indicates that ‘The RT-qPCR results matched the trends shown in the Nanostring data (Supplementary Fig. 9), indicating that our transcriptomic data was robust’, but there are some differences seen with the Nanostring results as shown in ‘c’ particularly for the PLX vs Veh in the transgenic. Is it appropriate to state that the results matched? How does this influence the interpretation of the changes (for non validated genes)?

We thank the reviewer for the question and have performed additional qPCR experiments to measure the five most highly expressed IEGs in our samples (*Btg2*, *Cyr61*, *Dusp1*, *Fos*, and *Nr4a1*). Consistent with our transcriptome analysis, we found a male-specific up-regulation of IEGs after PLX treatment (Supplementary Fig. 14a). Furthermore, all five genes showed robust correlation between expression level and brain PLX concentration (Supplementary Fig. 14c). Additionally, we further validated *Fos* expression by *in situ* labeling of mRNA in mouse brain sections using RNAscope (Fig. 6k,l). Based on these data, we are confident that up-regulation of IEGs after PLX treatment is a robust phenomenon. While we showed a general agreement between the transcriptome and RT-qPCR results, we agree with the reviewer that it is difficult to infer such consistency in genes lacking the qPCR data. Therefore, we toned down the language related to this conclusion.

1.3. I appreciate the authors’ explanation that doing gene expression analysis at end of life is not practical and variable, and understand the end point of 7 months for the analysis. However this should not preclude the requested comparison of female vs male gene expression from the Nanostring dataset – as it is there are 2 genes highlighted showing sex differences in Fig.7f. Are these the only 2 genes which were significantly different between transgenic mice + PLX in males vs females?

We appreciate the reviewer’s question. As the reviewer expected, there were many sex-specific expression patterns observed in our datasets. However, the primary focus of our study was not on intrinsic sex differences per se, but rather to understand the sex-specific repercussions of CSF1R inhibition. Figure 7f (now 6f) was meant to highlight such differences, and we showed, as examples, one gene highly correlated with PLX concentration (*Fos*), and one gene that correlated with sex independent of PLX (*Uty*). Our goal was to show that these two patterns were clearly different. To best represent all genes that may have such patterns, we presented this correlative data in Figure 6g. From this analysis, it became clear that many of the highly correlated genes were immediate early genes. We appreciate that there may be interest in other genes that we did not highlight in Figure 6, and thus we are glad to now provide Supplementary Data File 2 which shows the expression levels of all genes from both male and female PLX-treated mice, as well as a statistical comparison between the two sexes. Importantly, our entire Nanostring dataset which includes data from both brain regions, both Tg2541 and wild type mice, both sexes, and both PLX- and vehicle-treated mice is also available from Github and the link is provided in the “Code Availability” section. Thus anyone with interest in other sex-specific differences may perform those comparisons using our data.

As microglia are the primary, drug-targeted, cell type in our studies we performed deeper analyses of sex-specific gene expression profiles of surviving microglia in PLX-treated mice (Fig. 10). We found a greater upregulation of inflammation-related genes in male mice after PLX treatment (Fig. 10e-h). This finding is in agreement with our characterization of microglial

morphology (Fig. 9), all pointing to a male-specific inflammatory phenotype after PLX treatment. Furthermore, we validated sex-specific PLX-upregulated microglial genes by qPCR (Fig. 10f) and by *in situ* mRNA hybridization (RNAscope) analyses of mouse brain sections (Fig. 10g). We also performed mass spectrometry-based metabolomics analyses and found increased levels of the excitatory neurotransmitter glutamate (as well as the inhibitory neurotransmitter GABA) in the brains of male PLX-treated Tg2541 mice only (Fig. 7), indicative of neuronal excitotoxicity. We believe that these new analyses provide mechanistic insight explaining the sex-specific neurofunctional benefits in female mice and toxicity in male mice (see response to Reviewer 2).

1.4. For the brain NFL measurements, this does show a nice correlation between increasing concentration of PLX and increased brain NFL in males, but not females (new Fig.5i,j). The female NFL levels actually appear higher than the male ones in the hindbrain. Can the authors explain how this could mean that the males treated with PLX have increased neurotoxicity vs females?

This is a very astute observation and indeed, the levels of NfL in the hindbrains of female Tg2541 mice presented in Fig. 4j are higher than in male mice when grouped ($P = 0.0153$). However, there are several important considerations when interpreting this observation. Firstly, whereas NfL levels in plasma are a validated biomarker of neurodegeneration, it is entirely unclear what NfL levels in brain homogenates actually represent. The NfL measured in brain homogenates could be intracellular protein that was freed during homogenization/lysis, it could be extracellular protein that was released from neurons but not yet taken up by the vascular/glymphatic system, or it could derive from residual plasma in blood vessels (these tissues are harvested from non-perfused mice). As such, higher NfL levels in female mice could possibly indicate greater neuronal survival relative to male mice. Additionally, the data we present indicating neuronal excitotoxicity in male mice, namely increased brain expression of immediate early genes (IEGs) (Fig. 6h-l and Supplementary Fig. 14) and increased excitatory neurotransmitter levels (Fig. 7), is based on analysis of forebrain samples, not hindbrain. Indeed, we did not observe any difference in NfL levels in the forebrains of female Tg2541 mice presented in Fig. 4i relative to male mice when grouped ($P = 0.6145$). Therefore, we do not equate brain NfL with neurotoxicity, nor does our data necessarily indicate sex-dependent neuronal excitotoxicity in the Tg2541 mouse hindbrain.

1.5. With regards to the astrocyte signature, the field is moving away from using A1 and A2 terminology to avoid assumptions about function based on expression of a few genes. I suggest rephrasing to say ‘genes previously associated with neurotoxic astrocytes following LPS exposure or neuroprotective function in an artery occlusion model’, or something to that effect.

We agree with the reviewer that describing astrocyte function based only on a few genes is too simplistic, and we have corrected the text to tone down the interpretation of astrocytic function. While imperfect, we used the “PAN”, “A1” and “A2” labels in Fig. 8d with the purpose of selecting established subsets of astrocyte genes rather than broad analysis of all astrocyte genes, or only relying on GFAP. As the reviewer suggested, we have added the following sentence to the manuscript on Line 1341-1344: “Based on a prior study (Nature 26;541(7638):481-487, 2017), A1 genes are associated with neurotoxic astrocytes following lipopolysaccharide exposure and A2 genes are associated with neuroprotective function in an artery occlusion model”.

I have been asked to comment on responses to Reviewer 3’s issues as they are unable to re-review, and I’ve done this to the best of my ability below:

1.6. I disagree with Reviewer 3 that this paper is incremental as the impact of sex on differential response to CSF1R inhibition in pathology has not been reported to my knowledge, and is important to consider for the field not just for potential therapeutic applications but also for experimental use of CSF1R inhibitors in models of disease.

We thank the reviewer for recognizing the importance of our coherent scientific discovery of the impact of sex on the differential response to CSF1R inhibition in tauopathy.

1.7. I also disagree with the opinion on the inappropriate use of IBA1, as although this can pick up all macrophage populations in the CNS, the more microglia 'specific' markers with good antibodies available are downregulated in the context of chronic injury so are likely not going to work. This does come back to my issue on perivascular macrophage depletion though, so IBA1 will pick up presence and depletion of both populations, and this should be acknowledged in the paper.

We agree and have acknowledged that "Iba1 labels both PVMs and microglia" on Line 1516.

1.8. The reviewer asked whether there was an impact on metabolism in females vs males, and although the authors had previously implied differences in metabolism based on activity measurement, however metabolism was not directly measured. I can't comment on whether this is what the reviewer was getting at nor how this would best be measured. This is what I assumed from the question as the activity data was already included in the first submission.

We thank Reviewer 1 for their comment. In our very first submission we showed that male and female mice had similar food consumption (drug intake) rates (Supplementary Fig. 10a,b), yet male mice were significantly less active than female mice (Supplementary Fig. 10c,d; $P=0.0047$ for PLX3397, $P=0.0024$ for PLX5622), which is strong evidence that male mice have slower drug metabolism rates. Based on that data, we clearly stated in our first submission: "the reduced PLX exposure in female mice is likely due to a higher metabolic and drug clearance rate compared with male mice". We believe that Reviewer 3 simply missed that conclusion since they still asked "Is it due to slower metabolism of the drug?".

1.9. The reviewer raises the point that IEGs can go up with PLX treatment without concomitant increase in NFL. I accept the authors' answer that the presence or absence of tau likely influences the sensitivity of neurons to the PLX conditions.

We thank the reviewer for their comment.

1.10. The reviewer brings up microglia repopulation. The authors mention that previous papers showed that microglia repopulate to normal baseline levels, and this is when they decided to reinitiate the treatment. I think the rationale makes sense, but one must acknowledge that microglia repopulation in a pathological setting does not always lead to microglia coming back to baseline transcriptional profiles (eg. in aging), so this may have happened in the transgenic as well. This should be acknowledged in the paper in the discussion.

We agree and have added this concept to the Discussion section on Lines 609-612.

2. Reviewer #2:

The authors made impressive efforts to address the referee' comments. Yet, the manuscript still

suffers from being "patchy" rather than one coherent scientific discovery. In addition, it still misses a mechanistic explanation for the sex differences that is important and interesting but requires a more comprehensive mechanistic insight.

We thank the reviewer for noting our major advances in the manuscript. We also appreciate the reviewer's persistent opinion that the storyline in our manuscript was "patchy". We have taken this concern seriously. As you will see, with all of the new data and analyses presented in this round and the previous round of revision, we have achieved a cohesive scientific story that provides a mechanistic basis for our findings and conclusions. In addition to the substantial additions to the manuscript, we have also updated the title and abstract to reflect this complete mechanistic and preclinical study.

In order to gain further insight into the underlying mechanism related to sex-specific PLX-induced effect, we evaluated the gene expression patterns of residual microglia in male and female mice individually following PLX treatment. We found that PLX treatment led to the activation of a sub-cluster of microglial genes in male mice only. Interestingly, pathway analysis revealed that inflammation-related pathways were predominantly enriched in this male-specific activation pattern (Figure 10). Consistent with this male-specific inflammatory microglia gene expression pattern, we observed that the microglial morphology remained in an activated state following PLX treatment in male mice, whereas microglia female mice adopted a more ramified morphology associated with quiescent microglia similar to those found in wild type mice (updated Figure 9). Interestingly, these inflammatory phenotypes (Villa et al. 2018, Cell Rep; Hong et al. 2010, J Biol Chem) and morphological features (Vinet et al. 2012, J Neuroinflammation) of microglia have been previously linked with excitotoxicity, which provides a mechanistic basis for the male-specific drug-induced neurotoxicity and astrocytosis we discovered in the study (Figs. 6-8). To confirm this mechanism, we then performed mass spectrometry-based metabolomics analyses and found increased levels of the excitatory neurotransmitter glutamate (as well as the inhibitory neurotransmitter GABA) in the brains of male PLX-treated Tg2541 mice only (Fig. 7), indicative of neuronal excitotoxicity.

In conclusion, our new data complement the prior results showing that CSF1R inhibition must occur in the CNS (Fig. 5), and collectively provide a cohesive and mechanistic story: we found morphological and transcriptional changes in microglia associated with tau deposition, consistent with a pattern of pathological activation and associated neurotoxic astrocytes. CSF1R inhibition preferentially eliminated these microglia in female mice, leaving the brain with a more quiescent and less inflammatory microglial population, which led to neurofunctional rescue and extended survival. Male mice, on the other hand, developed a drug-induced inflammatory microglial phenotype, which contributed to neuronal excitotoxicity characterized by neurotransmitter dysregulation and a lack of neuroprotection or life extension. Our study is now a comprehensive preclinical characterization of CSF1R-antagonism for treating tauopathy, and will be invaluable for further development of this drug target in microglia.

REVIEWERS' COMMENTS

Reviewer #1 (Remarks to the Author):

The authors have addressed my issues with new data and analyses. I am satisfied with their response and have no further issues to raise.